# Stem cells repurpose proliferation to contain a breach in their niche barrier

Kenneth Lay[1], Shaopeng Yuan[1], Shiri Gur-Cohen[1], Yuxuan Miao[1], Tianxiao Han[1], Shruti Naik[1], H Amalia Pasolli[2], Samantha B Larsen[1], Elaine Fuchs[1]*

[1]Robin Neustein Laboratory of Mammalian Cell Biology and Development, Howard Hughes Medical Institute, The Rockefeller University, New York, United States; [2]Electron Microscopy Shared Resource, Howard Hughes Medical Institute, Janelia Research Campus, Virginia, United States

**Abstract** Adult stem cells are responsible for life-long tissue maintenance. They reside in and interact with specialized tissue microenvironments (niches). Using murine hair follicle as a model, we show that when junctional perturbations in the niche disrupt barrier function, adjacent stem cells dramatically change their transcriptome independent of bacterial invasion and become capable of directly signaling to and recruiting immune cells. Additionally, these stem cells elevate cell cycle transcripts which reduce their quiescence threshold, enabling them to selectively proliferate within this microenvironment of immune distress cues. However, rather than mobilizing to fuel new tissue regeneration, these ectopically proliferative stem cells remain within their niche to contain the breach. Together, our findings expose a potential communication relay system that operates from the niche to the stem cells to the immune system and back. The repurposing of proliferation by these stem cells patch the breached barrier, stoke the immune response and restore niche integrity.

DOI: https://doi.org/10.7554/eLife.41661.001

## Introduction

Adult stem cells reside in most if not all tissues. They are endowed with the ability to self-renew long-term and differentiate into specialized cell lineages in order to maintain, renew and repair tissues. In turn, stem cells are heavily influenced by the tissue microenvironment in which they reside. Together, the tight interaction and cooperativity between stem cells and their niche allow stem cells to perform their critical task of maintaining tissue homeostasis and function throughout the lifetime of the organism.

To date, most studies on the niche have focused on the various factors it secretes to control stem cell behavior during normal homeostasis. These factors include both positive and negative signals emanating from a diverse array of niche cells, including stromal cells, blood vessels, nerves and stem cell progeny (*Asada et al., 2017*; *Hsu et al., 2014a*; *Yu and Scadden, 2016*). Far less is known about how stem cells cope with perturbations in niche integrity and orchestrate responses to restore niche homeostasis. For epithelial tissue stem cells, this is particularly important, as they must respond quickly to and repair a breach in their epithelial barrier that excludes harmful microbes and retains essential fluids.

In the current study, we tackle this problem by taking advantage of the relatively simple architecture of the hair follicle (HF) bulge, the niche in which stem cells of the HF reside (*Cotsarelis et al., 1990*; *Hsu et al., 2014a*). The main part of the bulge is bilayered. It consists of a single concentric layer of stem cells that is sandwiched between a basement membrane on the outer side and a layer of terminally differentiated epithelial barrier 'niche' cells on the inner side. These 'inner bulge' niche cells are exposed to the external environment through an open channel that enables the hair to

*For correspondence:
fuchslb@rockefeller.edu

**eLife digest** Most, if not all, tissues of an adult animal contain stem cells. These stem cells regenerate and repair damaged tissues and organs for the entire lifetime of an animal, contributing to a healthy life. They divide to make daughter cells that become either new stem cells or specialized cells of that organ.

Adult stem cells exist in specific areas within tissues known as niches, where they interact with surrounding cells and molecules that inform their behavior. For example, cells and molecules within these niches can signal stem cells to remain in a 'dormant' state, but upon injury, they can mobilize stem cells to form new tissue and repair the wound. So far, it has been unclear how stem cells sense damage and stress and direct their efforts away from their normal duties towards repair.

Here, Lay et al. studied the stem cells in the mouse skin that are responsible to regenerate hair. Every hair follicle contains a niche (the 'bulge'), where these stem cells live and share their environment with cells that anchor the hair. The niche tethers to the stem cells through specific adhesion molecules that also help the niche to form a tight seal to prevent bacteria from entering. Lay et al. removed one of the adhesion molecules called E-cadherin, which caused a breach in the niche's barrier.

The stem cells sensed their damaged niche, prepared to multiply, and sent out stress signals to the immune system. The immune cells then arrived at the niche and sent signals back to the stem cells, prodding them to multiply and patch the barrier, while at the same time, keeping the inflammation in check. This remarkable ability of the stem cells to recruit immune cells and initiate a dialogue with them enabled the stem cells to divert their attention from regenerating hair and instead directing it towards the site of the tissue damage.

Other stem cells, such as those in the lung or gut, may have similar mechanisms to detect and respond to physical damage. It will be interesting to investigate the underlying mechanism of how immune cells are involved in balancing stem cell regenerative capacity and response to physical damage. A better knowledge of these processes could help to regenerate tissues or even entire organs.

DOI: https://doi.org/10.7554/eLife.41661.002

protrude from the skin surface (*Figure 1A*). The cells derive from the stem cells, but they do not do so directly within the bulge. Rather, the barrier cells are generated late in the terminal differentiation pathway of stem cell progeny that produce the new hair (*Hsu et al., 2011*).

Indeed, besides forming this epithelial barrier, the bulge stem cells also fuel the episodic bouts of hair follicle (HF) regeneration ('anagen') needed to replenish the protective hair coat of the animal. In between episodic cycles of hair growth, the bulge exists in a resting state ('telogen'), anchoring the hair made in the previous hair cycle (*Figure 1A*). At the bulge base ('hair germ'), quiescent stem cells interact with a specialized underlying mesenchymal structure, called the dermal papilla (DP). This cross-talk is necessary to yield a threshold of activating factors (BMP inhibitors, WNTs) that launch a new hair cycle (*Hsu et al., 2011*).

Stem cells tether to their inner bulge niche through cadherins, the core transmembrane components of adherens junctions (AJs), which coordinate intercellular adhesion, junctional integrity and proliferation in epithelial tissues (*Stepniak et al., 2009*; *Takeichi, 2014*). AJs also stabilize tight junctions, which prevent passage of bacteria and other entities across an epithelial sheet (*Rübsam et al., 2017*; *Tunggal et al., 2005*). While bulge stem cells express both E- and P-cadherin, we discovered that the inner bulge niche expresses only E-cadherin.

Here, we report that selective loss of E-cadherin within the inner bulge niche triggers a barrier breach that results in microbial invasion. We show that breached niche cells elevate an adhesive and cytoskeletal transcriptome. By contrast, the resident stem cells, which are normally quiescent, begin to proliferate in a fashion that we show is independent of microbial influence. Strikingly, they also activate a transcriptome that, when compared to proliferating anagen-phase stem cells in a normal niche, is similar in cell cycle genes, but differs in a cohort of immune-signaling genes. Moreover, although the cell cycle transcriptome lowers their threshold for quiescence, pushing them into proliferation requires this additional facet of the immune-signaling transcriptome. Indeed, the myriad of

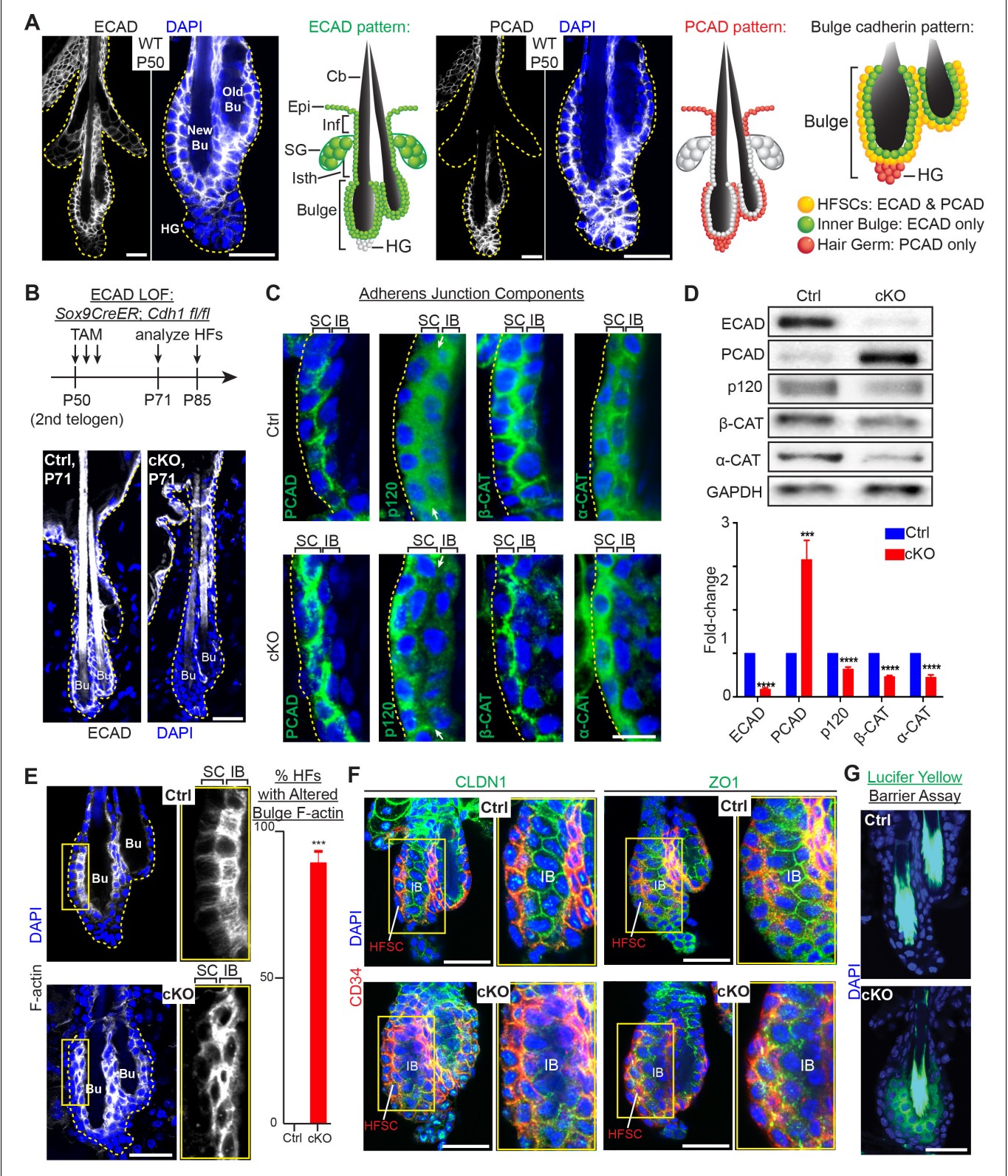

**Figure 1.** Adherens junctions in the niche but not its stem cells are affected by E-cadherin depletion. Images are from whole-mount immunofluorescence microscopy; DAPI (blue) marks chromatin. Antibodies are color-coded. (A) E-cadherin and P-cadherin expression (magnified views at right), summarized in drawings. Cb, club hair; Epi, epidermis; Inf, infundibulum; SG, sebaceous gland; Isth, isthmus; hair germ, hair germ; HFSCs, hair follicle stem cells. Scale bars, 30 μm. (B) Strategy of inducing *Cdh1* ablation during the extended 2nd telogen and analyzing thereafter. Images show
*Figure 1 continued on next page*

*Figure 1 continued*

effective E-cadherin depletion in cKO bulge and isthmus by postnatal day 71 (P71). Scale bar, 30 µm. (**C**) Bulge expression of AJ proteins P-cadherin, p120-catenin, β-catenin and α-catenin. Shown are magnified views of bulge bilayer, with outer layer of stem cells (SC) and inner layer of 'inner bulge' (IB) niche cells (see *Figure 1—figure supplement 2A* for zoomed out views). White arrows highlight the paucity of p120 at the *Cdh1* cKO stem cell: niche interface. Scale bar, 10 µm. (**D**) Immunoblots of AJ proteins. Data are mean ±SEM of≥4 independent replicates of FACS-purified bulge HF stem cells normalized to GAPDH. \*\*\*p < 0.001; \*\*\*\*p < 0.0001. (**E**) Phalloidin staining reveals perturbations in F-actin within the inner bulge (IB), arising from *Cdh1* ablation. Right, quantifications (n = 4 mice per condition/genotype; N = 20 HFs per mouse). Data are mean ±SEM. \*\*\*p < 0.001. (**F**) Whole-mount Z-stack imaging of tight junction components claudin one and zona occludens 1 (green). HF stem cells are co-labeled by CD34 (in red). Note paucity of tight junction labeling within the inner bulge (IB), arising from E-cadherin loss. (**G**) Barrier assay. Underlying dermis was removed from HFs and epidermis, which were then submerged in Lucifer yellow at 37°C for 3 hr, followed by fixation, mounting and imaging. Scale bar, 30 µm.

DOI: https://doi.org/10.7554/eLife.41661.003

The following figure supplements are available for figure 1:

**Figure supplement 1.** *Cdh3*-Null HFs have proper architecture and undergo hair cycling normally.

DOI: https://doi.org/10.7554/eLife.41661.004

**Figure supplement 2.** E-cadherin is critical to maintain the integrity of the stem cell niche.

DOI: https://doi.org/10.7554/eLife.41661.005

**Figure supplement 3.** Desmosomes are intact in E-cadherin-deficient bulge.

DOI: https://doi.org/10.7554/eLife.41661.006

chemo-attractants, cytokines and growth factors induced by these stem cells when they experience a damaged niche is required to recruit immune cells into this otherwise purportedly 'immune privileged' site (*Meyer et al., 2008*; *Paus and Bertolini, 2013*). The outcome of this atypical two-pronged path to stem cell proliferation, namely enhanced immune signaling and cell cycling, is a repurposing of new daughter stem cells from their normal task of tissue regeneration and towards patching the breached barrier.

Our findings have strong relevance to a breadth of human skin conditions where progenitor cell proliferation is coupled to barrier breach. Our data reveal that stem cells have the potential to sense defects in the integrity of their niche and respond by activating two branches of a transcriptional program, which together cope with the compromised tissue function surrounding them. Additionally and importantly, we show that the niche is not merely a source of secreted factors that control stem cell activity and tissue growth. Rather, it is a key structural entity whose integrity is closely monitored by its vigilant stem cell residents.

## Results

### E-cadherin is critical for adherens junction formation in the inner bulge niche but not in the stem cells

We became interested in the possibility that cadherins may function critically in HF stem cell niche integrity when we analyzed bulge cadherin expression during the extended, synchronized telogen phase of the second hair cycle in mice (*Figure 1A*). E-cadherin localized to most epithelial cell-cell boundaries, displaying the strongest immuno-labeling at the interface between niche cells and stem cells. Conversely, P-cadherin specifically concentrated at stem cell intercellular junctions and was absent from niche cells. This was consistent with the ~20X higher *Cdh3* transcript level in the stem cells versus niche cells. Indeed, as judged by enzyme-linked immunosorbent assays (ELISAs) on protein lysates of bulge stem cells [purified by fluorescence-activated cell sorting (FACS) of skin cell suspensions], P-cadherin levels were even higher than E-cadherin (*Figure 1—figure supplement 1A and B*). As expected from our expression data and the functional redundancy of these cadherins (*Tinkle et al., 2008*), P-cadherin (*Cdh3*) null mice (*Radice et al., 1997*) displayed a normal hair coat and hair cycle (*Figure 1—figure supplement 1C*). E-cadherin was maintained at all cell-cell junctions within the P-cadherin depleted bulge, and hair germ cells became proliferative on cue during early anagen (*Figure 1—figure supplement 1D*). To explore the functional consequences of E-cadherin loss of function (loss of function) in the bulge, we conditionally ablated its gene (*Cdh1*) in *Cdh1*$^{fl/fl}$; *Rosa26*$^{lox-STOP-lox-YFP}$ reporter mice by using a tamoxifen (TAM)-inducible CreER knocked into the endogenous locus of *Sox9*. By inducing *Cdh1* ablation near the beginning of 2$^{nd}$ telogen (postnatal

day P50), E-cadherin was efficiently depleted throughout the bulge when analyzed 3 weeks later (*Figure 1B*).

*Cdh1*-null HF stem cells continued to display AJs along stem cell-stem cell borders, reflective of P-cadherin compensation (*Figure 1C* and *Figure 1—figure supplement 2A and B*). In fact, by immunoblot analyses and quantifications of proteins from FACS-purified $\alpha6^+CD34^+$ HF stem cells, P-cadherin levels in E-cadherin deficient HF stem cells were even higher than normal (*Figure 1D* and *Figure 1—figure supplement 2C*), a feature confirmed by knocking down *Cdh1* transcripts through shRNA (*Figure 1—figure supplement 2D*).

By contrast, the *Cdh1*-null inner bulge layer lacked P-cadherin and in turn AJs on all cell-cell borders (*Figure 1C*). Since AJ formation requires intercellular cadherin-cadherin binding, this also left the stem cell-niche interface devoid of AJs. This was best exemplified by the paucity of p120-catenin at this interface (arrows in *Figure 1C*). Notably, p120-catenin binds directly to classical cadherins and is required for AJ stabilization. Immunoblot analyses confirmed the overall reduction in these other AJ proteins, despite upregulation of P-cadherin in stem cells within the *Cdh1*-depleted bulge (*Figure 1D*).

## E-cadherin is critical for maintaining the integrity of the stem cell niche

AJs form an intercellular network of F-actin across an epithelial sheet of keratinocytes (*Vaezi et al., 2002*; *Vasioukhin et al., 2000*). In wild-type HFs, F-actin organization was particularly robust within the inner bulge, suggesting a special role for the cytoskeletal-intercellular networks within this layer of niche cells. However, in the absence of E-cadherin, the actin cytoskeletal organization was markedly perturbed throughout the inner bulge, including the stem cell-niche interface (*Figure 1E*).

Probing deeper, we next examined tight junctions. As shown in *Figure 1F* and *Figure 1—figure supplement 2E*, the expression, localization, continuity and stability of tight junction proteins were largely intact in the CD34+ stem cells, but disrupted in the niche cells of the bulge. Moreover, when we tested for tight junction integrity by exposing HFs to Lucifer yellow, the dye penetrated the inner bulge layer deficient for E-cadherin (*Figure 1G*).

Although our findings underscored the importance of E-cadherin in maintaining AJs, tight junctions and the integrity of the niche, niche cells were not lost from the bulge: desmosomes were still present in seemingly normal morphology, numbers and localization, and intercellular membranes at the niche-stem cell interface were still sealed (*Figure 1—figure supplement 3A and B*). The inner bulge layer also remain adhered to the club hair via 'half-desmosomal' structures as previously described for WT bulge (*Hsu et al., 2011*) (*Figure 1—figure supplement 3C*).

## Loss of niche integrity results in precocious stem cell proliferation without initiating hair growth

As in the normal hair cycle, telogen-phase stem cells in *Cdh1*-heterozygous HFs were quiescent until anagen onset, when their proliferation initiated first within the hair germ (AnaI), and several days later within the bulge (AnaII-III; *Figure 2A*, first three panels) (*Hsu et al., 2014b*). Upon homozygous *Cdh1* ablation, however, telogen-phase bulge stem cell residents began proliferating (*Figure 2A*, fourth panel; quantifications at right). In striking contrast to the normal hair cycle, this was neither preceded nor accompanied by hair germ proliferation.

Normally, whether in homeostasis or in response to a wound, for example hair plucking, HF stem cell proliferation results in progeny that exit their niche and launch a new round of hair cycling (*Hsu et al., 2011*; *Morris et al., 2004*; *Tumbar et al., 2004*). By contrast, the daughters of proliferative events in the telogen *Cdh1*-null bulge remained within the niche (*Figure 2A*). Moreover, these HFs actually spent longer than normal in telogen prior to entering a new round of hair growth (*Figure 2B*). These results indicated that the normal telogen phase cues were still intact, which we later document further.

Instead of being utilized to launch a new hair cycle, this telogen-phase stem cell proliferation produced additional bulge layers, readily evident by 5 weeks post-*Cdh1* ablation (*Figure 2C*). Daughters retained stem cell identity as judged by bulge stem cell markers CD34 and LHX2, and by factors such as TCF4 and SOX9, which are expressed but not exclusively by HF stem cells (*Figure 2D* and *Figure 2—figure supplement 1A*). Quantifications revealed a significant expansion of stem cell numbers within the *Cdh1*-null bulge (*Figure 2D*).

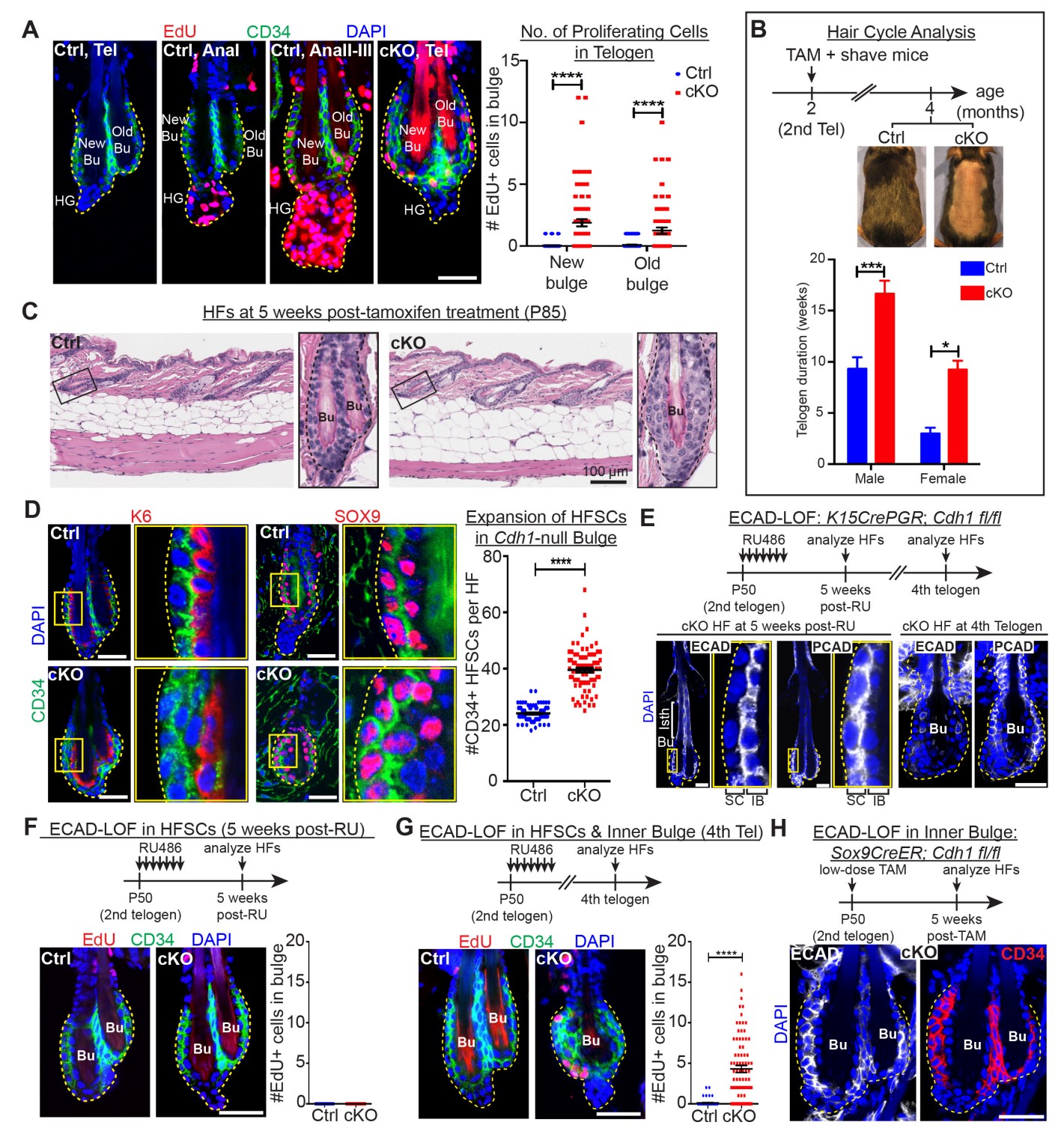

**Figure 2.** Telogen stem cells proliferate when E-cadherin is depleted from the bulge. Scale bars, 30 μm unless indicated otherwise. (**A**) (Left) Whole-mount immunofluorescence of HFs from mice pulsed with nucleotide analogue EdU for 24 hr prior to analysis. Proliferation dynamics are compared for control (Ctrl) HF, telogen (Tel) vs anagen (Ana) sub-stages I-III, and *Cdh1* cKO HF in telogen. (Right) Quantifications of EdU+ CD34+ HF stem cells (n = 4 mice per condition/genotype; N = 20 HFs per mouse). Data are mean ±SEM. ****p < 0.0001. (**B**) Top: Following *Cdh1* ablation in the bulge of 2$^{nd}$ telogen HFs, mice were shaved and monitored for hair coat recovery. Within 2 months, Ctrl mice had regenerated a new hair coat and entered their 3$^{rd}$ telogen, but *Cdh1* cKO HFs were still in their 2$^{nd}$ telogen. Bottom: Telogen duration was determined as the period between week 8 of age (when

*Figure 2 continued on next page*

*Figure 2 continued*

TAM was administered) and the week during which 50% of shaved back skin had entered anagen (n = 4 mice per condition/genotype). Data are mean ± SEM. *p < 0.05; ***p < 0.001. (C) Skin histology, as examined by hematoxylin and eosin staining. Magnified views of boxed areas are shown, with black dashed lines denoting epithelial-dermal boundaries. Note hyperthickening of *Cdh1* cKO bulge, compared to Ctrl. (D) *Cdh1* bulge expansion derives from stem cells, as revealed by immunofluorescence for markers of stem cells (CD34+, SOX9+) vs inner bulge niche cells (K6+, SOX9+). Yellow-boxed regions are magnified at right of images. Quantifications of HF stem cell (HFSC) numbers at right (n = 4 mice per condition/genotype; N = 20 HFs per mouse). Data are mean ±SEM. ****p < 0.0001. (E–G) HF stem cell-specific ECAD loss of function was induced by activating *Krt15CrePGR* in 2nd telogen. HFs were analyzed by whole-mount immunofluorescence in 2nd or 4th telogen post-ablation (see *Figure 2—figure supplement 1D* for more details). (E) In 2nd telogen, ECAD is specifically depleted in HF stem cells (SC; boxed regions are magnified at the right). Following hair cycling, the inner bulge (IB) derives from HF stem cell progeny and hence is also *Cdh1*-null by 4th telogen. Hyperthickened bulge architecture correlates with ECAD loss of function in the niche cells of the inner bulge. (F–G) EdU labeling performed 24 hr prior to analyses shows that ECAD loss of function in HF stem cells alone does not elicit telogen-phase proliferation (F), while ECAD loss of function in both HF stem cells and inner bulge causes marked proliferative response (G). Data are mean ±SEM (n = 4 mice per condition/genotype; N = 20 HFs per mouse). (H) Inner bulge-specific ECAD loss of function was induced as indicated (see *Figure 2—figure supplement 1E* for more details). Note expansion in surrounding WT CD34+ stem cells, not seen in HF stem cell-specific ECAD loss of function.

DOI: https://doi.org/10.7554/eLife.41661.007

The following figure supplement is available for figure 2:

**Figure supplement 1.** Dissecting the causes of telogen HF stem cell proliferation upon loss of E-cadherin.

DOI: https://doi.org/10.7554/eLife.41661.008

Despite the increased proliferation in telogen, *Cdh1*-null HF stem cells were still sensitive to the activating cues of the hair cycle, and returned to a non-dividing state as inhibitory levels rose in subsequent anagen and telogen (*Figure 2—figure supplement 1B*). This further suggested that the normal cues that govern hair cycling were still in place.

## Wild-type stem cells proliferate when residing in a compromised niche

We next addressed whether the bulge stem cells proliferated because of their own *Cdh1* loss or that of the niche. Since changes in AJs can impact proliferation (*Stepniak et al., 2009*), we first considered whether the elevated P-cadherin seen in the *Cdh1*-null stem cells might be involved. However, when we engineered mice whose skin epithelium was selectively transduced with doxycycline (Doxy)-inducible *Cdh3*, we saw no significant effect on stem cell proliferation upon induction of P-cadherin over-expression (*Figure 2—figure supplement 1C*).

Next, we used *Krt15-CrePGR* mice (*Morris et al., 2004*) to selectively ablate *Cdh1* in bulge stem cells during the extended 2nd telogen (*Figure 2E* and *Figure 2—figure supplement 1D*). Strikingly, the ectopic stem cell proliferation and expansion of bulge layers that we had observed upon *Cdh1* ablation in both stem cells and inner bulge, did not occur upon stem cell-specific ablation (*Figure 2F*). These data pointed to perturbations within the niche as the source of the stimulus for neighboring stem cells to proliferate.

Since the inner bulge niche is derived from stem cell progeny that terminally differentiate towards the end of the hair cycle (*Hsu et al., 2011*), we could test our hypothesis by monitoring *Krt15CrePGR*-activated *Cdh1* cKO mice through subsequent hair cycles, when E-cadherin was also lost in the niche (*Figure 2E*). Indeed, bulge stem cells proliferated and expanded concomitant with E-cadherin loss within the inner bulge (*Figure 2G*).

Finally, by titrating down the TAM dosage administered to *Sox9CreER; Cdh1*$^{fl/fl}$*; Rosa26*$^{lox-STOP-lox-YFP}$ mice, we restricted Cre activity to the inner bulge layer (*Hsu et al., 2011*) (*Figure 2—figure supplement 1E*). Notably, selective *Cdh1* ablation in the inner bulge niche was sufficient to elicit an architectural shift to a triple layered bulge, even though the stem cells were now WT (*Figure 2H*). Together, these findings suggest that bulge stem cell proliferation may rely at least in part upon the direct interface between the stem cells and their defective inner bulge niche.

## Bacteria infiltrate the breached niche barrier but are not required for stem cell proliferation

The inability of the *Cdh1* null inner bulge to retain dye (*Figure 1G*) suggested that its integrity is needed to provide a protective barrier to the external environment. Upon testing this possibility, we found that in contrast to controls, *Cdh1* cKO HFs were infiltrated with microbes, as shown both by

Gram-positive (deep violet) and Gram-negative (pink) bacterial stains and by fluorescence in situ hybridization (FISH) for a pan-bacterial 16S rRNA (*Figure 3A and B*). These findings highlighted the importance of a healthy inner bulge niche in guarding its stem cells and the skin against microbial infiltration.

A priori, the stem cells might be responding to these microbes that infiltrate the breached barrier, rather than the damaged niche per se. To test for the impact of these infiltrating bacteria, we treated mice with antibiotics to deplete bacteria from the skin. As judged by in situ hybridization for 16S ribosomal RNA, microbes were efficiently thwarted (*Figure 3B*). Despite efficient microbial clearing, the *Cdh1* cKO bulge still remained proliferative and hyper-thickened (*Figure 3C*). Altogether,

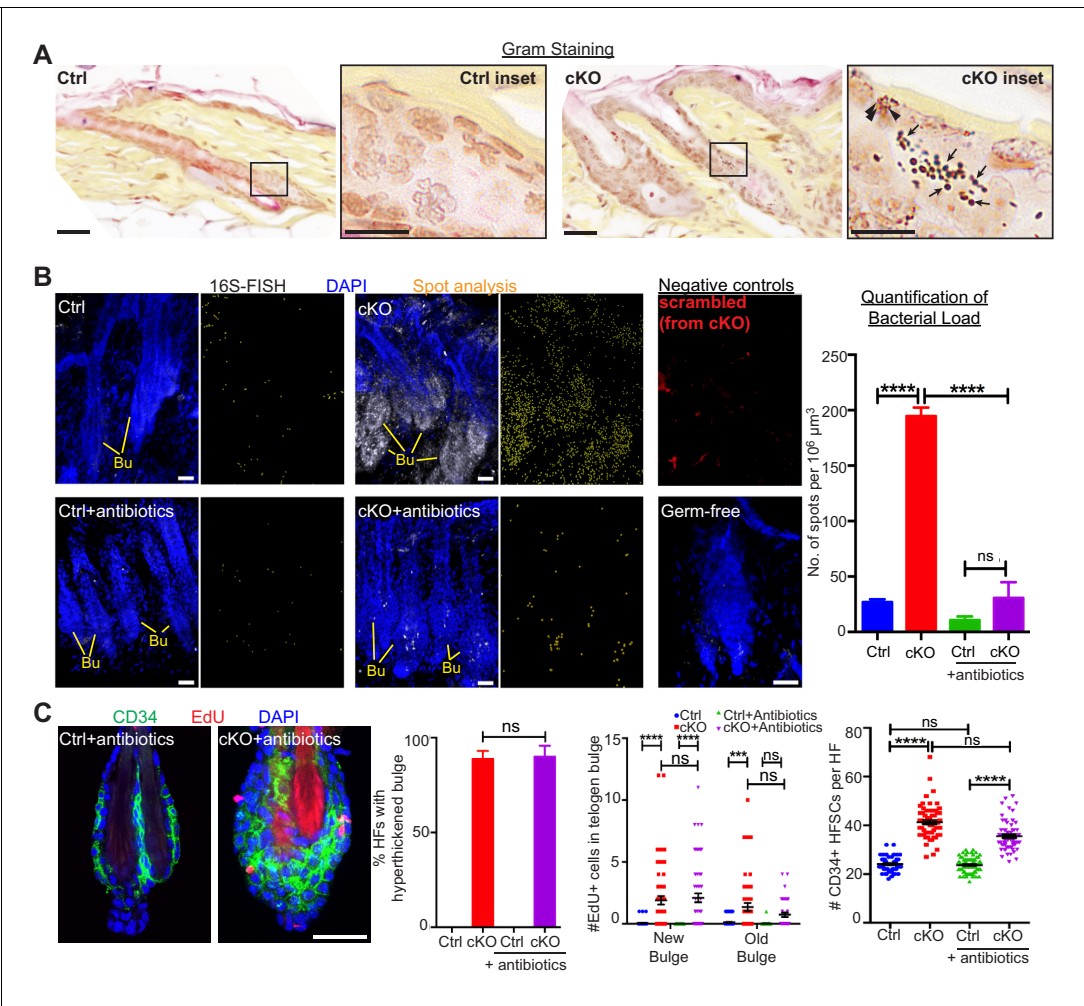

**Figure 3.** When the stem cell niche is breached, bacteria infiltrate but are not required for the stem cell response. (**A**) Gram staining of sagittal skin sections reveals near absence of bacteria in control HFs but presence of Gram-positive (deep violet; arrows) and Gram-negative (pink; arrowheads) bacteria in *Cdh1* cKO HFs. Scale bars, 30 μm. Magnified images are of boxed areas (scale bars,10 μm). (**B**) Fluorescence in situ hybridization (FISH) of pan-bacterial 16S rRNA (white) in cleared skin whole-mounts, co-labeled for DAPI (blue). Spot analysis of 16S-FISH signal (yellow) was used to quantify bacterial load per $10^6$ μm³ of skin. Negative controls included simultaneous FISH with scrambled probe on all samples (shown here in red from cKO), and analyses on germ-free mice (in bottom right frame). Antibiotics treatment of *Sox9CreER; Cdh1^{fl/fl}* cKO mice depleted bacterial load to near control levels, as quantified at right. Scale bars, 30 μm. Data are mean ±SEM (n = 3 mice per condition/genotype). ****p < 0.0001; ns, non-significant. (**C**) Whole-mount immunofluorescence of telogen HFs from *Cdh1* cKO and control mice treated with antibiotics. Note persistence of multiple layers of CD34+ proliferative stem cells within *Cdh1* null HF bulge despite microbial depletion. Accompanying graphs quantify changes in niche architecture, EdU incorporation and HF stem cell (HFSC) numbers, presented as mean ±SEM (n = 3 mice per condition/genotype; N = 20 HFs per mouse). ***p < 0.001 ****p < 0.0001; ns, non-significant. Scale bar = 30 μm. *Sox9CreER; Cdh1^{fl/fl}* cKO and control (without antibiotics) datasets are from *Figure 2*.

DOI: https://doi.org/10.7554/eLife.41661.009

while our findings do not discount a role for bacterial signals, they rule them out as the instigating factor in causing stem cell proliferation when the niche barrier was breached.

## Immune cells are recruited to the breached niche independently of bacteria

Upon barrier breach, a swarm of immune cells, positive for the pan-leukocyte marker CD45, selectively surrounded the bulge (*Figure 4A*). Although CD45+ numbers per unit area of skin varied, when we quantified the number of CD45+ cells within a 25 μm radius of the bulge, we consistently observed a marked and statistically significant increase in immune cells surrounding the *Cdh1* null bulge as compared to the WT control bulge (*Figure 4C*, *Cdh1*-null bulge [Sox9CreER] vs control).

Given the microbial invasion, an immune response was expected. Yet surprisingly, we found that immune cells had already begun to accumulate by D23 post *Cdh1* ablation, before the increase in 16S rRNA and shift to a triple layered bulge architecture which were robust only by D35 (*Figures 2C* and *4B*). Moreover, when we depleted bacteria with antibiotics, immune cells were still recruited to the breached niche (*Figure 4C* and *Figure 4—figure supplement 1A*). By contrast, when we used *Krt15CrePGR*-activated mice to selectively ablate *Cdh1* from 2nd telogen HF stem cells, immune cells were not recruited until subsequent hair cycles, when the inner bulge layer became *Cdh1*-null (*Figure 4C* and *Figure 4—figure supplement 1B and C*). A robust immune response also encircled the bulge when we ablated *Cdh1* selectively in the inner bulge and not in stem cells (*Figure 4C* and *Figure 4—figure supplement 1D*).

While many immune cells, including dendritic cells (DCs), macrophages, γδ dendritic epidermal T cells (DETCs), γδ TCR+ dermal T cells, and αβTCR+ dermal T cells, are known to be residents of the skin (*Naik et al., 2015*; *Scharschmidt et al., 2017*; *Scharschmidt et al., 2015*), this immune response arising from the breached niche went well beyond the normal surveillance status of the immune system. Moreover, the composition of the CD45+ immune cells that infiltrated the breached bulge showed clear differences from the normal resident immune cell patterns. Most notable was the increase in immunostaining for CD3+ T cells and CD11b+ myeloid lineage cells, including MHC-class II+ dendritic cells and F4/80+ macrophages (*Figure 4D and E* and *Figure 4—figure supplement 1E*). To further characterize, distinguish and quantify these cell types, we used these and additional markers to analyze immune cells by flow cytometry (*Figure 4D–4F* and *Figure 4—figure supplement 1F and G*).

We observed a significant increase in the MHCII+ CD11c+ cells (*Figure 4D*) which are defined as dendritic cells since these cells are also negative/low for CD64, Mertk or Ly6C (*Figure 4—figure supplement 1F*). Additionally, although CD64+ Mertk+ macrophage numbers were more variable, and hence not statistically significant, they did follow a trend of being higher than normal, and preferentially associating with the breached niche (*Figure 4D* and *Figure 4—figure supplement 1E*).

Focusing on T cells that were concentrated near breached bulge niches, γδ T cell and DETC numbers were comparable between WT and *Cdh1* cKO skins, although DETC numbers were quite variable even among control samples (*Figure 4E*). By contrast, αβ T cells showed marked and statistically significant increases in *Cdh1* cKO skins (*Figure 4E*). Many of these T cells were positive for FOXP3, a marker for regulatory T cells (Tregs), and further quantification revealed statistically significant increases in both absolute numbers and percentages of Tregs relative to the total αβ T cell population (*Figure 4F* and *Figure 4—figure supplement 1G*). Moreover, Treg numbers surrounding the bulge were markedly higher than the normal skin resident population (*Figure 4F*).

## Stem cells within a breached niche upregulate chemokine genes that recruit immune cells

Our data thus far indicated that the ability to elicit an immune response was independent of microbial presence, but dependent upon the barrier breach within the niche. A priori, this could mean that *Cdh1*-null inner bulge cells produce the chemokines that recruit the immune cells. Alternatively, upon detecting an AJ-deficient interface with the breached niche, stem cells, whether *Cdh1*-null or wild-type, might respond by producing and transmitting stress signals to the immune system.

To distinguish between these possibilities, we FACS-purified and transcriptionally profiled inner bulge niche cells (*Figure 5—figure supplement 1A*) and stem cells (*Figure 1—figure supplement 2C*) from telogen-phase *Cdh*1 cKO and control littermate bulges. To tease out changes specific to

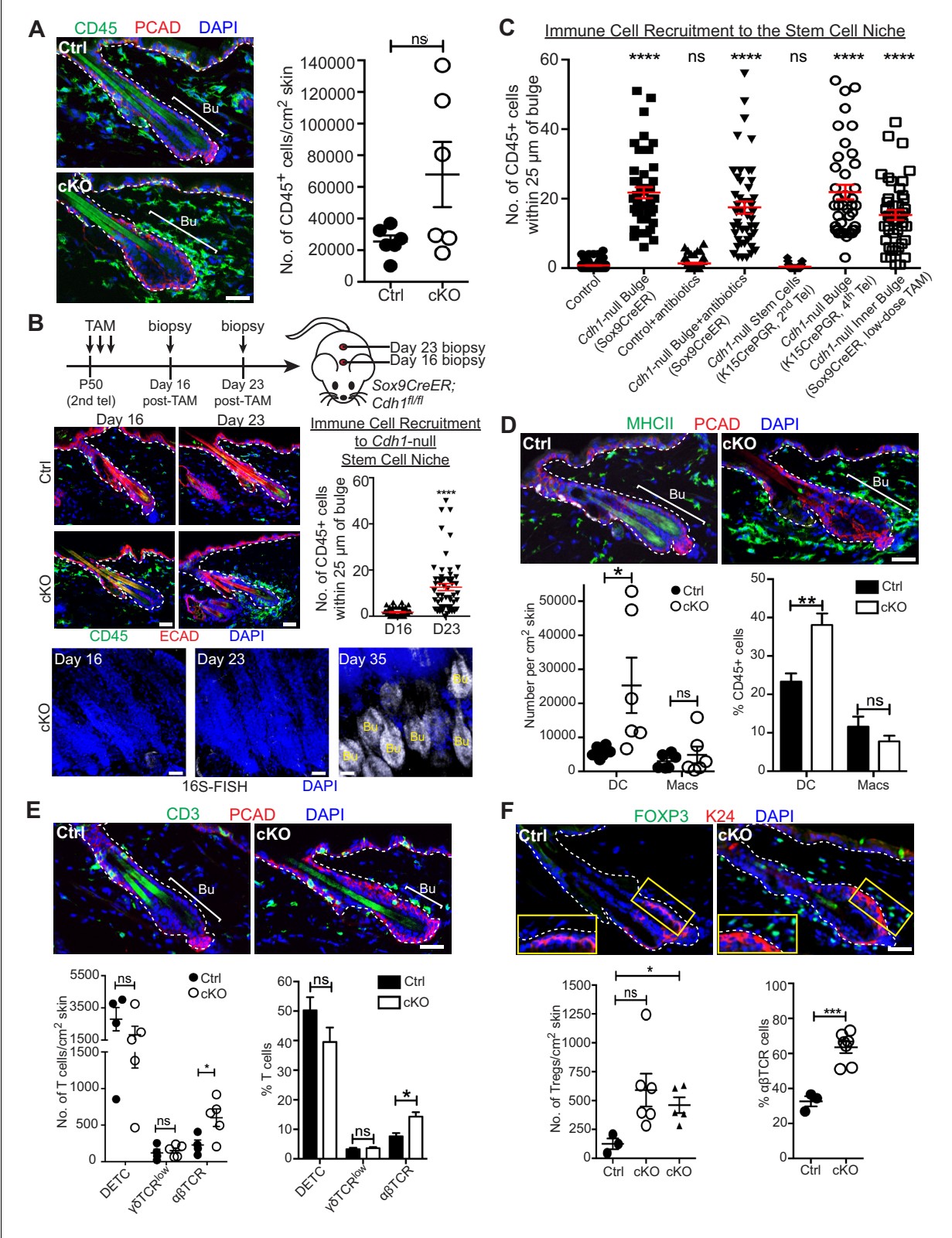

**Figure 4.** A barrier breach within the stem cell niche results in immune cell recruitment. Scale bars, 30 μm. (**A**) Immunofluorescence (left) and flow cytometry quantifications (right; details below) of immune cells (CD45+) in *Sox9CreER; Cdh1^{fl/fl}* cKO and control skins. (**B**) Sagittal skin sections were analyzed as indicated to determine the temporal relation between efficient E-cadherin depletion, recruitment of CD45+ immune cells and increase in 16S-FISH signals. Shown are quantifications of CD45+ immune cell numbers within a 25 μm radius from the bulge. Data are ±SEM (n = 4 mice per

*Figure 4 continued on next page*

*Figure 4 continued*

condition/genotype, N ≥ 10 HFs per mouse). ****p < 0.0001. (**C**) Quantifications of CD45+ immune cell numbers within a 25 μm radius from the bulge in sagittal skin sections from mice whose a) ECAD status within the bulge of telogen HFs, and b) status of antibiotic treatment, are shown. Data are ±SEM (n ≥ 3 mice per condition/genotype, N ≥ 10 HFs per mouse). In each case, significance testing was performed relative to control. ns, non-significant; ****p < 0.0001. Note that the swarm of CD45+ cells surrounding bulge is dependent upon ECAD loss of function in the niche not its stem cells. (**D–F**) Immunofluorescence and flow cytometry quantifications to characterize the CD45+ immune cell populations in *Sox9CreER; Cdh1^{fl/fl}* cKO and control skins. Antibodies used were against: MHC Class II [dendritic cells (DCs), monocytes and macrophages (Macs)]; CD11c (DCs); CD11b and CD64 (Macs); CD3 (T cells); γδTCR [dendritic epidermal T cells (DETCs), some dermal T cells]; TCRβ (αβ T cells, including regulatory T cells [Tregs]); and FOXP3 (Tregs). Note accumulation of immune cells, particularly DCs and Tregs, specifically around the *Cdh1*-null HF bulge. Flow cytometry data are mean ±SEM (n = 4–6 mice per condition/genotype as indicated), confirming the significant differences (*p < 0.05; **p < 0.01, ***p < 0.001) in αβ TCR cells, Tregs and DCs between Ctrl and cKO (ns, non-significant). [Note that for Treg quantifications in F, the open-circle dataset included an outlier that was consistent with our findings, but which made the difference statistically insignificant. The triangle dataset excluded the outlier, in which case the difference between cKO and control was statistically significant.] See *Figure 4—figure supplement 1A–E* for more immunofluorescence images. See *Figure 4—figure supplement 1F and G* for flow cytometry strategies used to perform quantifications here and in (**A**).

DOI: https://doi.org/10.7554/eLife.41661.010

The following figure supplement is available for figure 4:

**Figure supplement 1.** A barrier breach within the stem cell niche results in immune cell recruitment.

DOI: https://doi.org/10.7554/eLife.41661.011

stem cells residing in the breached niche, we also performed RNA-sequencing (RNA-seq) on WT bulge stem cells in anagen II/III, that is the only time when they naturally proliferate (*Figure 5—figure supplement 1B*). Our data were consistent across independent replicates, enabling us to generate molecular signatures of transcripts that were differentially expressed (p-value<0.05, absolute fold-change [FC]≥1.5) in either niche cells (1070↑, 1058↓) or stem cells (1194↑, 793↓) from *Cdh1*-null versus control bulges (*Figure 5—figure supplement 1C*).

Scrutinizing the specific genes within each of the top four KEGG pathway categories of the *Cdh1*-null inner bulge profile, we found that transcripts encoding cell adhesion, proteoglycans and cytoskeletal-associated proteins were featured highly, consistent with a feedback attempt by these junction-compromised cells to restore the barrier, while metabolic pathways were downregulated (*Figure 5A* and *Figure 5—figure supplement 1D*). By contrast, cell cycle genes and immune signaling and response genes dominated the most highly changed pathways in the stem cells from *Cdh1*-null bulge niches (*Figure 5B–5D*, upper Venn diagram and *Figure 5—figure supplement 1E and F*). Moreover, in contrast to these proliferative stem cells within the telogen *Cdh1*-null niche, neither proliferative WT stem cells within an Ana II/III WT bulge nor quiescent WT stem cells within a telogen WT bulge showed expression of these immune pathway genes (*Figure 5—figure supplement 1G*). Principal Component Analyses further highlighted these marked distinctions between proliferative bulge stem cells devoted to normal tissue growth and those repurposed here to add cell layers to the damaged niche (*Figure 5E*).

Our analyses also revealed salient differences in the transcriptional responses of niche versus stem cells within the damaged bulge (*Figure 5D*, bottom Venn diagram). This was true even for categories with some apparent overlap, such as cytokine:cytokine receptor interaction transcripts (*Figure 5A and B*). Indeed, closer inspection of individual genes within this and related immune cell categories of the niche transcriptome revealed that levels of these niche transcripts were often considerably lower and less significant than in their stem cell neighbors (*Figure 5F*). By contrast, of the cytokine and chemokine genes that were highly changed in the stem cell transcriptome (*Figure 5C*), most were substantially more highly expressed by these stem cells than their *Cdh1*-null niche neighbors, even when the stem cells surrounding a *Cdh1*-null inner bulge niche were wild-type (*Figure 5G*).

*Ccl1* and *Ccl2* were the top two most highly upregulated genes in this stem cell cohort (184X, 38X; *Figure 5C*), and like most of this transcriptome, were barely expressed by the neighboring breached niche. Notably, CCL1 and CCL2 are established chemokines for both DCs and macrophages (*Cantor and Haskins, 2007*; *Gombert et al., 2005*; *Li and Tai, 2013*; *Moser et al., 2004*). Importantly, not only were *Ccl1, Ccl2* and many other features of these signatures robustly expressed by the HF stem cells experiencing a breached niche, but this response still occurred even

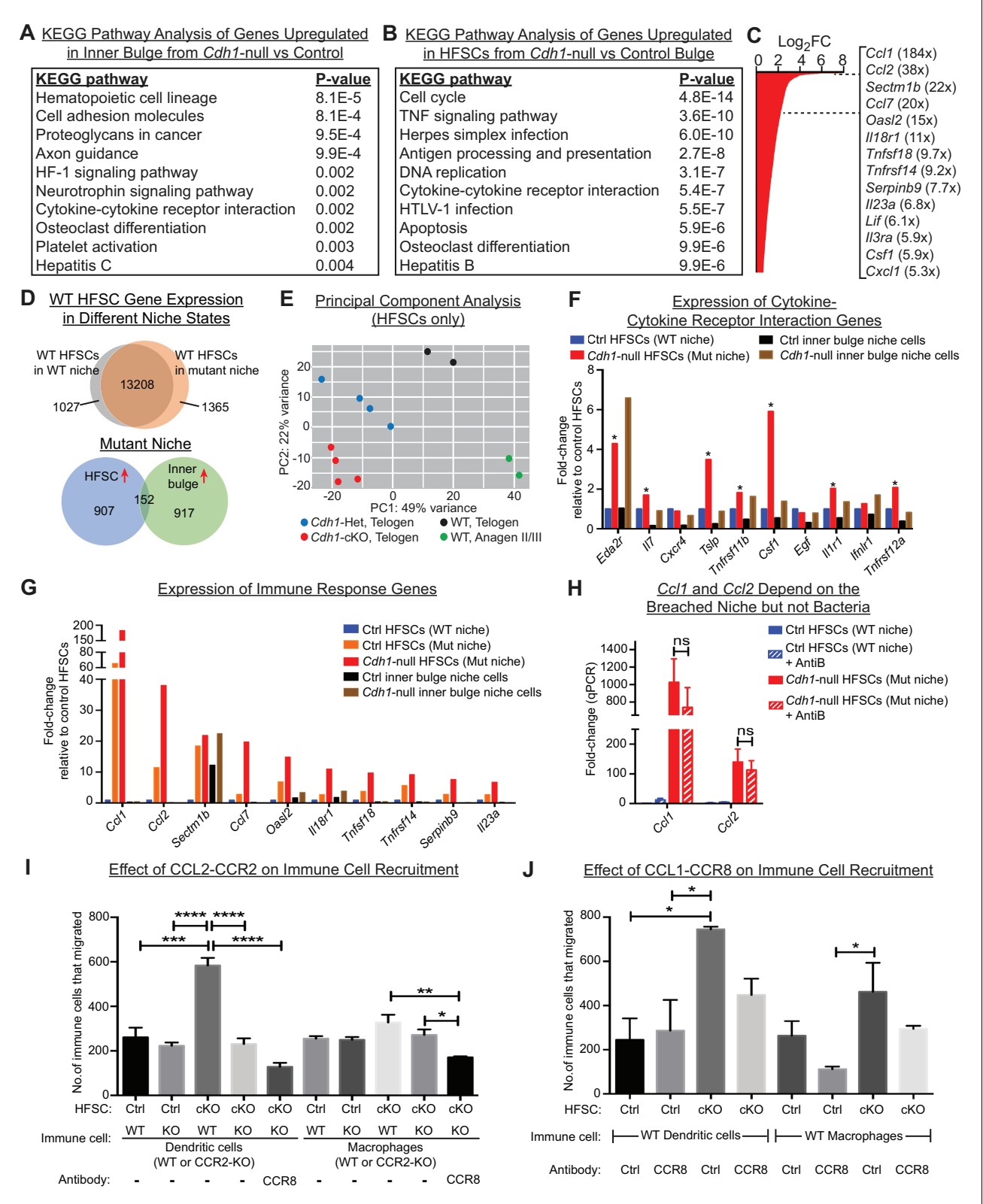

**Figure 5.** HF stem cells up-regulate chemokines to recruit immune cells. Stem cells or inner bulge niche cells were isolated from *Sox9CreER* cKO or matched littermate heterozygous control animals, that were either treated with low-dose Tamoxifen (Tam^low, to target only the inner bulge niche cells for *Cdh1* loss-of-function) or regular-dose Tamoxifen (Tam^reg, to target both inner bulge niche and stem cells). (A) KEGG pathway analyses of genes significantly up-regulated (p < 0.05, absolute fold-change ≥1.5) in *Cdh1*-null vs control inner bulge niche cells (both with normal stem cells). Shown are

Figure 5 continued

the top 10 of 30 pathways with p-values<0.05. (B) KEGG pathway analyses of genes significantly up-regulated (p < 0.05, absolute fold-change ≥1.5) in *Cdh1*-null vs control HF stem cells (HFSCs) from *Cdh1*-null or control bulge. Shown are the top 10 of 50 pathways with p-values<0.05. Overall, p-values are much lower for stem cells than inner bulge niche cells (A). (C) Fold-changes (FC) of significantly up-regulated genes in *Cdh1*-null stem cells from *Cdh1*-null bulge. Listed at right are top genes associated with immune response (FCs indicated in parentheses). (D) Top, Venn diagram comparing changed transcripts in WT stem cells that surround either a *Cdh1*-null mutant inner bulge niche (Tam^low) or a WT inner bulge niche (no Tam). Both populations commonly express 13208 genes (TPM [transcripts per kilobase million]≥1), but WT stem cells surrounding a mutant inner bulge gain expression of 1365 genes by ≥1.5 fold and lose expression of 1027 genes (TPM <1). Bottom, Venn diagram comparing gene expression changes in stem cells vs inner bulge niche cells from *Cdh1*-null bulge. Note that the stem cells and their niche adopt highly distinct transcriptomes (upregulated genes shown here). (E) Principal component analysis (PCA) comparing transcriptomes of telogen-phase HF stem cells of WT, *Cdh1* cKO and Het bulges, and of anagen II/III HF stem cells from WT bulges. (F, G) Relative expression of genes within the KEGG pathway term 'cytokine-cytokine receptor interaction' and genes associated with immune response that scored as significantly up-regulated (*Cdh1*-null vs control) for niche (F) or stem cells (G). Note that of the few immune signaling genes in the breached niche signature, most are more highly and significantly expressed by the stem cells (*) (F); conversely, many immune response genes in the stem cell signature are barely expressed, if at all, in the breached inner bulge niche (G). Navy and red bars correspond to data from Tam^reg mice; orange, black and brown bars correspond to data from Tam^low mice. (H) Two highly upregulated immune response stem cell genes, *Ccl1* and *Ccl2*, are still as highly expressed even after antibiotics treatment, underscoring their independence from bacterial infiltration. qRT-PCR data are normalized to *Ppib*. Data are mean ±SEM (n = 4 mice per condition/genotype). ns, non-significant. All data were from Tam^reg mice. (I, J) Transwell migration assays. *Cdh1*-null and Ctrl HF stem cells were seeded in bottom Boyden chamber. Bone marrow-derived dendritic cells (BMDCs) or macrophages (BMDM), either WT or deficient for CCR2 receptor, were placed in upper chamber, with or without isotype control or CCR8 blocking antibody. CD45+ BMDCs and BMDM that were chemo-attracted to bottom chambers were quantified by flow cytometry. Data are mean ±SEM (n = 3). Comparisons that are statistically significant are denoted by asterisks. *p < 0.05; **p < 0.01; ***p < 0.001; ****p < 0.0001.

DOI: https://doi.org/10.7554/eLife.41661.012

The following figure supplement is available for figure 5:

**Figure supplement 1.** HF stem cells from *Cdh1*-null bulge display a gene expression profile distinct from WT HF stem cells that are naturally proliferating in anagen.
DOI: https://doi.org/10.7554/eLife.41661.013

when microbial infiltration was repressed (**Figure 5H**). This suggested that the stem cells recruited immune cells in direct response to the damaged niche rather than the bacteria that infiltrated.

Since CCL2 is an established chemokine for the CCR2 receptor, expressed by both DCs and macrophages, we focused first on the potential interactions between stem cells and these immune cells. To mimic the damaged niche microenvironment in vitro, we used *Cdh1*-null HF stem cells, which retained elevated *Ccl2* in culture (**Figure 5—figure supplement 1H**). We then tested their ability against WT HF stem cells to recruit bone marrow-derived DCs and macrophages in transwell migration assays. As shown in **Figure 5I,** a significant increase occurred in chemotaxis of bone marrow-derived DCs. Moreover, this difference was abolished when we used DCs that lacked CCR2. On the other hand, while the migration of macrophages in response to *Cdh1*-null HF stem cells followed a similar trend, the difference was highly variable and thus not statistically significant. Strikingly, these in vitro patterns of DC and macrophage migration were consistent with our in vivo immune cell profiling (**Figure 4D**), and revealed that *Cdh1*-null but not WT stem cells can directly attract DCs, and to a lesser extent macrophages, through a CCL2-CCR2-dependent mechanism.

Conversely, CCL1 stimulates CCR8 receptors, which are on the surface of DCs and macrophages. To test the significance of HF stem cell CCL1, we repeated the cell migration assays as above, this time using a CCR8 blocking antibody to disrupt a potential CCL1-CCR8 interaction. Interestingly, while the reduction in chemotactic response of DCs and macrophages was not as robust as blocking the CCL2-CCR2 axis (**Figure 5J**), treatment of *CCR2*-null DCs or macrophages with CCR8 blocking antibody to simultaneously block both CCL1 and CCL2-mediated chemotaxis further reduced their migration to statistically significant levels (**Figure 5I**). These findings were important as they indicated that HF stem cells experiencing a barrier breach in their niche can activate chemokines and directly recruit the immune cells that we see elevated upon a barrier breach. Why DC recruitment in vivo appeared to be more consistent than macrophage recruitment is not clear. However, it is notable that DCs and macrophages respond to similar chemokines during recruitment, and depending upon the particular cytokine milieu, differentiation of inflammatory monocytes can be skewed to DCs or macrophages (**Geissmann et al., 2010**).

## Stem cell proliferation and bulge hyperthickening arise from immune cell stimulation

We next addressed whether immune cells are required to trigger the stem cell proliferation that generated a thickened bulge. We waited until D18 post-Tam treatment, when *Cdh1* was ablated, but immune cells had not yet been recruited to the bulge (*Figure 4B*). We then administered the immunosuppressant dexamethasone (Dex) daily for ~2.5 weeks, and analyzed mice. At this time, untreated counterparts were swarmed by CD45+ immune cells and the bulge was hyperthickened (*Figure 6A*).

While not eliminating the resident skin immune cells, Dex effectively blocked the immune infiltration that otherwise surrounded the breached niche barrier in *Cdh1* cKO HFs (*Figure 6A*). Significantly, Dex-treated *Cdh1* cKO HFs also displayed normal bulge architecture (*Figure 6B*). Accordingly, both the telogen-phase proliferation displayed by stem cells in their deficient niche, and the concomitant increase in stem cell numbers, were now abolished. This was further reflected by the enhanced BMP-signaling effector phosphorylated-SMAD1 (*Figure 6B*), which in normal stem cells only diminishes in Anagen II/III when the natural activating cues override BMP quiescence signals from the inner bulge niche (*Hsu et al., 2014b*). That said, even in the absence of Dex, there were still signs that the normal telogen signals were intact in the *Cdh1*-null bulge. Thus, inner bulge-derived BMP6, a prerequisite for these pSMAD1 signaling dynamics, was still expressed by the *Cdh1*-null inner bulge cells, and correspondingly, the HF stem cells experiencing the breached niche still expressed BMP target genes, including *Nfatc1* and *Foxc1* (*Figure 6—figure supplement 1A*).

Since ablating only one immune component can have a profound effect on the remaining immune cell repertoire, it was necessary to use total immune cell suppression, rather than genetic ablation of specific immune cell populations. Although Dex impacts full-anagen phase HFs by inducing their early entry into the destructive phase (*Kwack et al., 2017*), our study focused on telogen, where no adverse effects of Dex had been reported. However, it was important to assess whether it was suppression of the immune response or Dex itself that was impacting stem cell proliferation.

As shown in the growth curves in *Figure 6B*, Dex alone did not inhibit HF stem cell proliferation directly. Moreover, bacteria were still present in immunosuppressed skin, arguing against their involvement in the proliferation dynamics observed here (*Figure 6—figure supplement 1B*). This finding also added to the evidence that the microbial infiltration arose from the loss of junctional integrity within the *Cdh1*-null niche, and not from the associated architectural disturbances caused by stem cell proliferation in the *Cdh1*-null niche.

Given these collective results, we focused on immune cell infiltration as the root of stem cell proliferation and expanded bulge structure. Thus far, our immune analyses indicated that DCs and Tregs were elevated prominently and specifically around the breached barrier (*Figure 4*). Closer inspection revealed that DCs and Tregs were closely associated with each other when surrounding the breached niche, and Dex treatment markedly blocked their infiltration into the *Cdh1*-null bulge niche (*Figure 6C*). Moreover, the immunosuppressive effects of Dex appeared to be specific for the immune infiltration to the bulge, as immune cells elsewhere in the skin remained during the Dex treatment (*Figure 6C*). In this regard, the Dex-treated *Cdh1*-cKO pattern of CD3+ and MHCII + immune cells resembled that of untreated control skin (*Figure 6—figure supplement 1C*).

Since Dex blocked immune infiltration into the breached niche, and since immune infiltration was essential for the specific proliferation of bulge stem cells that we observed, we turned to in vitro studies to address whether DCs or Tregs might directly affect HF stem cell proliferation. For this purpose, we purified and activated DCs as before (*Figure 5I and J*), and employed established protocols (see Materials and methods) to stimulate freshly isolated splenic naïve CD4+ T cells with CD3/CD28 antibodies (conventional T cells; Tconv). For Treg induction, we also added TGFβ (*Fantini et al., 2007*). By compartmentalizing the immune cells, we then assessed the effects of their secreted factors on the proliferation of HF stem cells. As shown in *Figure 6D*, while DCs showed no appreciable effect on HF stem cells, Tregs appeared to be particularly robust in soliciting a proliferative response. In wild-type skin, resident Tregs, which are positive for NOTCH ligand JAG1, have been proposed to promote HF stem cell proliferation through direct engagement and activation of Notch target genes (*Ali et al., 2017*). If so, we might have expected to see an elevation of Notch target genes in the stem cells of the *Cdh1* null bulge, given the marked elevation of Tregs within close proximity. That said, the most striking canonical NOTCH target gene showing sensitivity to

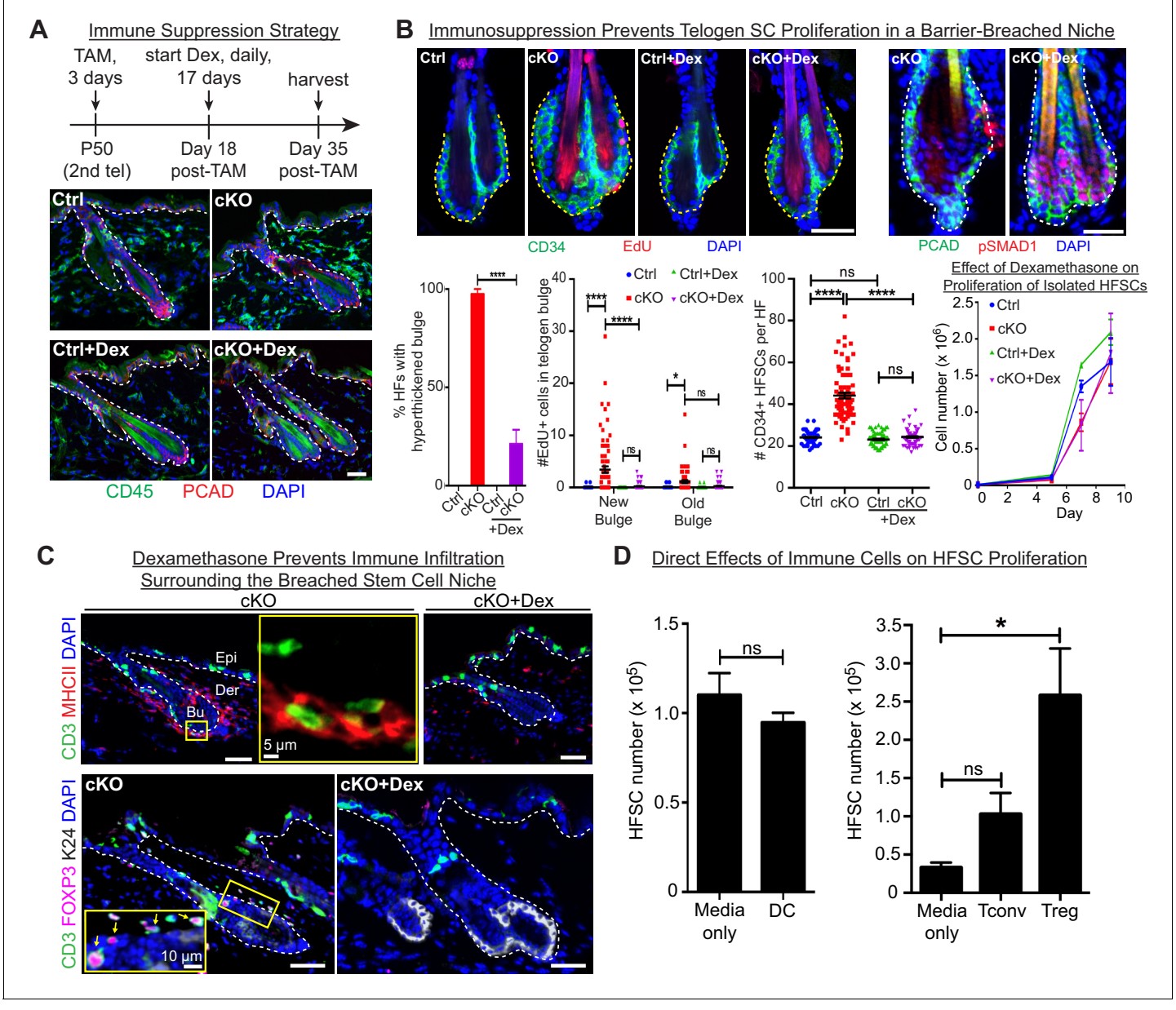

**Figure 6.** Stem cell proliferation and bulge hyperthickening arise from immune cell stimulation. Scale bars, 30 μm unless indicated otherwise. (**A**) Dexamethasone suppressed infiltration of immune cells to the bulge when administered after ECAD loss of function but prior to the immune response,. (**B**) As shown by CD34+, EdU incorporation and pSMAD1 whole-mount immunofluorescence, telogen-phase *Cdh1*-null bulge stem cells maintain quiescence in Dex-treated mice. Note that only when stem cell-stimulatory signals override BMP signals coming from the niche is pSMAD1 lost. Accompanying graphs quantify changes in niche architecture, HF stem cell (HFSC) proliferation and stem cell number, presented as mean ±SEM (n = 4 mice per condition/genotype; N = 20 HFs per mouse). ns, non-significant; *p < 0.05; ****p < 0.0001. Graph at right shows that Dex does not significantly impact HF stem cell proliferation in vitro; data are mean ±SEM (n = 3). (**C**) Upon Dex treatment, the influx of DCs and Tregs to the breached stem cell niche is blocked. Note that resident immune cells still show similar patterns. Note also the close proximity of DCs and Tregs in the cKO bulge region (magnified in top middle panel). (**D**) Tregs directly impact stem cell proliferation, as shown by in vitro co-culture assays of FACS-purified *Cdh1* cKO HF stem cells with BMDCs, conventional T cells (Tconv) or FACS-purified Tregs, prepared and tested as described in the methods. Experiments were performed in triplicate. In contrast to Tconv and DCs, the effects of Tregs were substantial (p < 0.05). Data are mean ±SEM. ns, non-significant; *p < 0.05.

DOI: https://doi.org/10.7554/eLife.41661.014

The following figure supplements are available for figure 6:

**Figure supplement 1.** Analysis of BMP signaling in HF stem cells and inner bulge niche cells with breached barrier function.

DOI: https://doi.org/10.7554/eLife.41661.015

*Figure 6 continued on next page*

*Figure 6 continued*

**Figure supplement 2.** E-cadherin expression in the HF stem cell bulge niche is essential to maintain tissue barrier function.

DOI: https://doi.org/10.7554/eLife.41661.016

dexamethasone was *Hey1,* and in this case, transcripts were up, rather than down when the Treg swarm was diminished (*Figure 6—figure supplement 2A*). We also examined the status of a cohort of HF stem cell genes reported to bind activated NOTCH in a ChIP-seq analysis of a muscle cell line (*Castel et al., 2013*) (*Figure 6—figure supplement 2B*). Indeed, most of these and established NOTCH target genes were very lowly expressed in normal telogen phase HF stem cells, consistent with the well-documented role for NOTCH signaling in HF differentiation, rather than bulge stem cell proliferation (*Demehri and Kopan, 2009*). Importantly, NOTCH target genes were not elevated appreciably in HF stem cells when the niche barrier was altered and swarmed with Tregs. Rather, we surmise that Tregs' proliferative effects that we observed on HF stem cells here are more likely to be rooted in amphiregulin and/or keratinocyte growth factor, known to be robustly expressed by Tregs in an inflammatory setting (*Arpaia et al., 2015*; *Dial et al., 2017*).

Similarly, in some contexts, the tension sensor YAP, is retained by α-catenin in the cytoplasm and becomes nuclear upon a loss of adherens junctions (*Li et al., 2016*; *Neto et al., 2018*; *Schlegelmilch et al., 2011*), leading to a marked increase in target genes of the YAP/TEAD transcription factor complex (*Liu and Wang, 2015*; *Zanconato et al., 2015*; *Zhang et al., 2011*). Since the telogen-phase HF stem cells in a *Cdh1*-mutant niche were proliferative, and YAP is known to influence proliferation, the elevation in *Ccne1* and other YAP target genes governing proliferation seemed at first glance to suggest altered YAP activity. However, Dex treatment largely normalized these cell cycle-related differences, and most other YAP target genes showed no major changes in expression irrespective of immune suppression (*Figure 6—figure supplement 2C*). In the absence of expression changes attributable to *Cdh1* status, and given the multiple alternative and possibly confounding means by which YAP is known to be regulated (*Deng et al., 2018*; *Totaro et al., 2018*), we did not pursue this further.

## The stem cell signature depends upon the breached barrier

Finally, to identify the portion of the stem cell transcriptome signature that is independent of immune cell infiltration, we performed RNA-seq on stem cells isolated from Dex-treated *Cdh1*-null and control bulges of telogen-phase HFs and compared their transcriptomes to their non-treated counterparts. A total of 509 genes were found to be significantly upregulated in stem cells from Dex-treated *Cdh1*-null versus control niches. Of these, 392 genes (77%) were also upregulated in their non-treated counterparts (*Figure 7A*).

Despite stem cells being un-proliferative under Dex treatment, the top KEGG pathway enriched among these 392 overlapping genes was still cell cycle (*Figure 7B*). Although the expression of these cell cycle transcripts was dampened by immunosuppressive conditions (*Figure 7C*, purple vs. red bars), they were still significantly upregulated, independent of both Dex treatment (*Figure 7C*, purple vs. green bars) and *Cdh1* loss (*Figure 7C*, orange vs. blue bars). A similar trend was observed for many of the immune response genes, whose expression persisted in HF stem cells in a breached niche even under immunosuppressive conditions (*Figure 7D*). Taken together, our findings suggested that the necessary threshold level of cell cycle transcripts required to push telogen-phase stem cells into proliferation was contributed in part by the *Cdh1*-null niche and in part by the ability of these stem cells to recruit immune cells.

Seeking why telogen-phase proliferation of stem cells was dependent upon not only immune cells but also niche status, we were drawn once again to the reduction in adherens junction proteins that stem cells experienced when they resided in the *Cdh1*-null bulge niche (*Figure 1C and D*). An inverse correlation between adherens junctions and cell proliferation is well-established (*Stepniak et al., 2009*), and our findings here pointed to a potential link at the transcriptional level.

Of additional note, many of the immune response transcripts that remained elevated in the HF stem cells of Dex-treated cKO skin were putative NFkB target genes, including *Tnfa, Ccl2, Ccl1, Csf1, Cxcl1* and *Ccl20* (*Figure 7E*). Previously, we showed that when p120-catenin (*Ctnnd1*) is conditionally ablated in skin epithelia, IkB kinase (IKK) activity is elevated, resulting in the phosphorylation

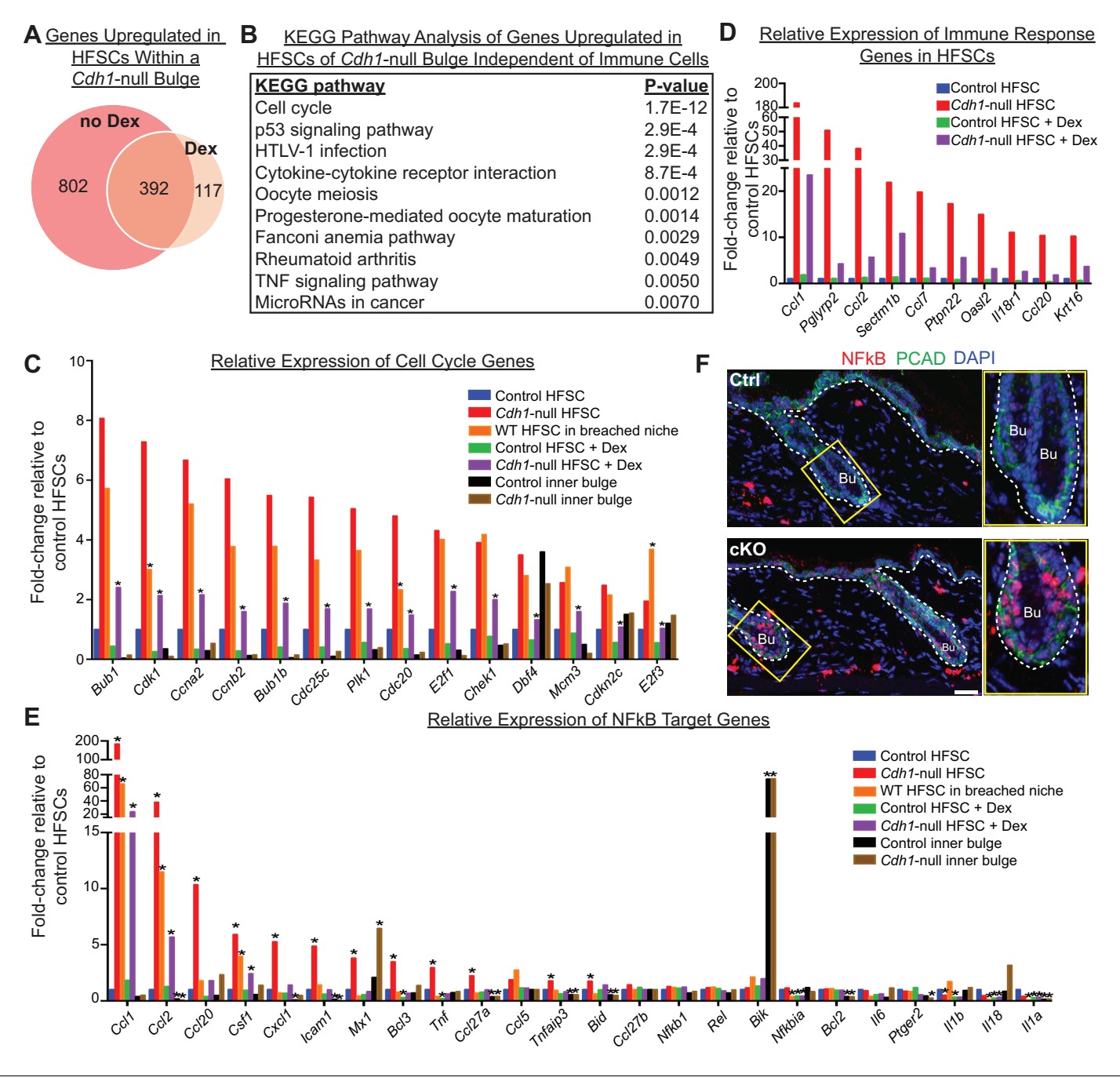

**Figure 7.** Stem cell transcriptomic changes occur in direct response to a barrier breach. (A, B) Comparisons of transcriptomes of HF stem cells (HFSCs) from *Cdh1* cKO vs.Ctrl bulges ±Dex. Venn Diagram (A) showed 392 overlapping genes that were significantly up-regulated (p < 0.05, fold-change [FC]≥ 1.5, 509 genes total) independent of Dex. (B) KEGG pathway analysis of these genes. (C) Relative expression of 'Cell cycle' genes from (B) in various HF stem cell and inner bulge populations. These genes are also significantly upregulated (p < 0.05) in WT SCs in a breached niche vs. control SCs. *p < 0.05 relative to *Cdh1*-null HFSCs. (D) Relative expression of immune response genes in stem cells from *Cdh1*-null bulge vs. Ctrl bulge in the absence or presence of Dex treatment. (E) Relative expression of NFkB target genes in various HF stem cell and inner bulge populations. *p < 0.05 (with absolute FC ≥1.5) relative to control HF stem cells. (F) Immunofluorescence for phosphorylated p65, a subunit of the NFkB transcription factor. Note increased nuclear p-p65 signal in *Cdh1*-cKO bulge (Bu) indicative of NFkB signaling activity.

DOI: https://doi.org/10.7554/eLife.41661.017

of IkBα and the release of the transcription factor NFkB into the nucleus (*Perez-Moreno et al., 2006*). Intriguingly, by immunofluorescence, NFkB showed nuclear localization in telogen-phase *Cdh1*-null HF stem cells (*Figure 7F*), which also had reduced p120 levels (*Figure 1C and D*). Although dexamethasone has been reported to dampen NFkB signaling by elevating IkBα transcription (*Auphan et al., 1995*; *Scheinman et al., 1995*), it only had a partial effect in *Cdh1*-null HF stem cells under conditions where IkBα phosphorylation was enhanced (*Figure 7E*). While further details are beyond the scope of the present study, our results suggest a mechanism whereby when the barrier is breached, reductions in adherens junctions impacts the activity of transcription factors such as NFkB, that can directly alter the transcriptional landscape of the HF stem cells in a way that promotes immune cell recruitment, and reduces, but does not overcome, the threshold for proliferation (*Figure 8* and *Figure 6—figure supplement 2D*).

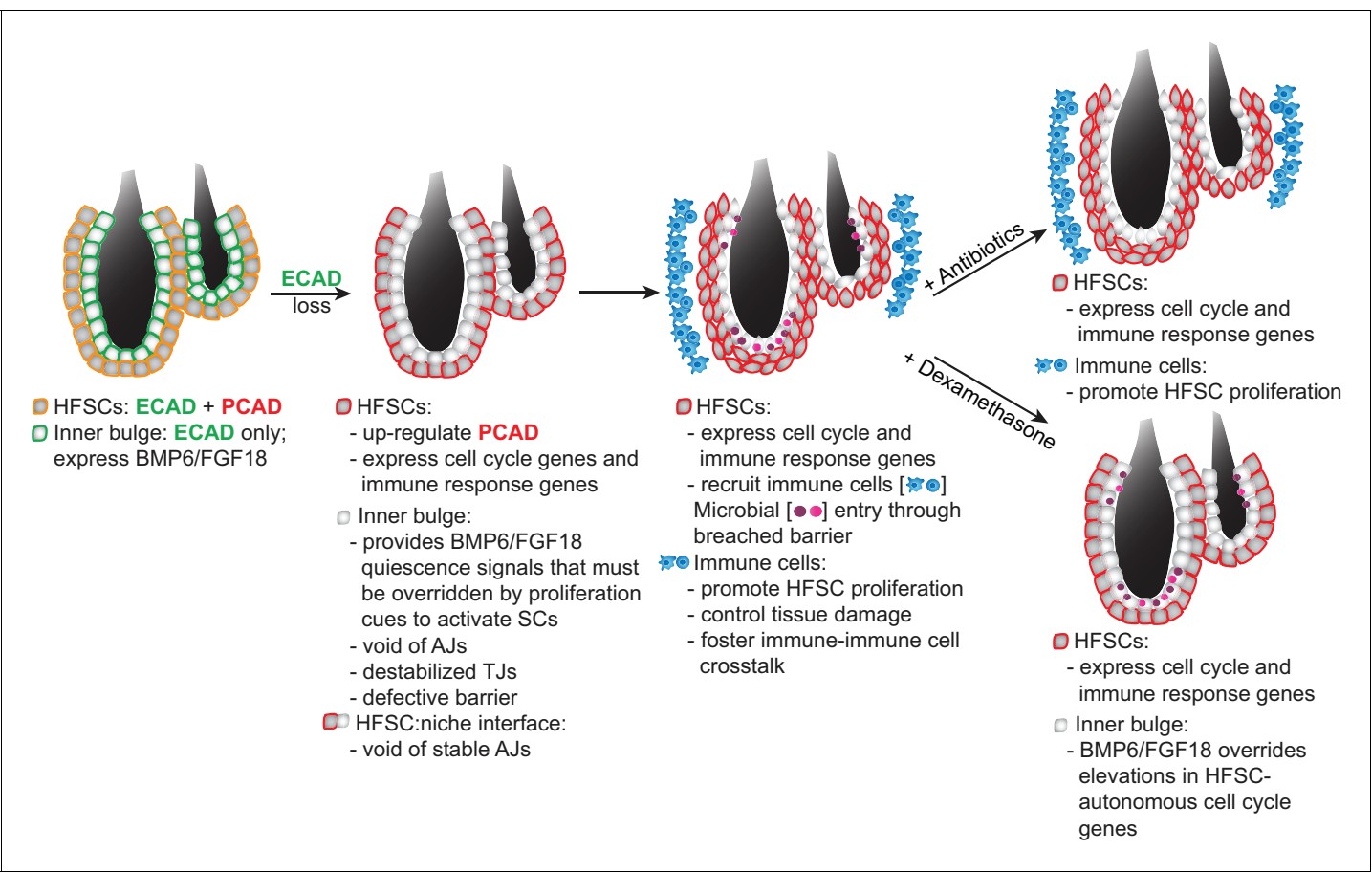

**Figure 8.** Model Summarizing How Stem Cells Respond to a Breach in Their Niche Barrier and Launch a Patch Process by Repurposing Stem Cell Proliferation. During telogen, inner bulge niche cells express BMP6 and FGF18, which maintain stem cell quiescence. When the sole cadherin (ECAD) of the inner bulge is removed, the niche cannot make adherens junctions (AJs), leading to tight junction destabilization, a barrier breach and microbial infiltration. Even though stem cells have compensatory PCAD and maintain stem cell-stem cell AJs, they cannot form AJs along the niche-stem cell interface. This reduction in AJ components induces a cell cycle transcriptome which lowers the threshold for proliferation, but is kept in check by niche inhibitory signals. Within their breached niche, stem cells also induce a transcriptome geared towards recruiting immune cells to the bulge, independent of the increased bacterial load. The stem cells need this altered microenvironment to tip their proliferative balance. Without anagen cues, newly generated stem cells remain within the bulge, where they are repurposed to boost the output of distress signals to the immune system and contain the breached barrier by reinforcing the bulge with layers of immune cells and stem cells that display intact cellular junctions.
DOI: https://doi.org/10.7554/eLife.41661.018

## Discussion

### The bulge niche functions as a barrier to protect its stem cells

The most important function of the skin epithelium is to provide the barrier that excludes harmful microbes and retains body fluids. When the barrier is breached, it must be repaired expeditiously to restore fitness to the animal. HFs extend deep into the skin, leaving an entry site for pathogens. The bulge niche of HF stem cells is strategically positioned at the base of this orifice that opens to the external environment. In our current study, we show that in addition to its role as a source of inhibitory signals for its stem cells (*Hsu et al., 2011*), the inner bulge provides the barrier that protects stem cells from microbial invasion.

In exploring how stem cells cope when their barrier is breached, we learned that by co-expressing P- and E-cadherin, the stem cells themselves are protected from the many situations where E-cadherin levels are naturally downregulated, which includes stem cell activation at the start of the hair cycle (*Lay et al., 2016*). Our new data substantiates this further, as when E-cadherin is lost from HF stem cells, they markedly upregulate P-cadherin.

### Stem cells respond to a breach in their niche's integrity by communicating it to the immune system

Through their immediate proximity to the inner bulge layer, the HF stem cells are poised to sense a barrier breach. Remarkably, our studies revealed that neither bacteria nor immune cells were required for this sensing to occur. Although the precise details await further investigation, the roots are likely to reside in the inability of stem cells to form adherens junctions with their *Cdh1*-null niche neighbors, which lack a cadherin backup system. As our data revealed, intercellular cadherin-cadherin junctions are featured prominently at the stem cell-niche interface of healthy HF bulges. Such junctions are known to act as key signaling centers and integrators of proliferation, polarity and the actin cytoskeleton across an epithelial tissue (*Padmanabhan et al., 2015*; *Stepniak et al., 2009*), and our current findings add to the mounting evidence that these junctions also serve as key regulators of inflammatory sensors (*Perez-Moreno et al., 2006*).

As we learned, once stem cells face a breach in their niche barrier, they trigger a distress transcriptional program that we show occurs even in the face of antibiotics or immune suppressive drugs and is strikingly different from the response elicited in the neighboring niche cells. Our findings are interesting in light of prior studies showing that under a number of different conditions, keratinocytes can express cytokines and chemokines that harbor potential to communicate with the immune system in normal homeostasis and in injury (*Adachi et al., 2015*; *Ali et al., 2017*; *Chen et al., 2015*; *Pasparakis et al., 2002*; *Zhang et al., 2004*). Of additional relevance to our study are previous observations that hair plucking enhances hair cycling, an effect which has been postulated on the one hand to be triggered by the loss of the BMP-expressing inner bulge niche cells (*Hsu et al., 2011*) but on the other hand to an elevation of *Ccl2* and inflammatory macrophages in the skin (*Chen et al., 2015*).

While intriguing, none of these prior studies have revealed whether stem cells are specifically capable of altering their transcriptional program in response to perturbations in their microenvironment and if they can, whether this endows them with the ability to adjust their behavior by directly recruiting immune cells. Notably, *Ccl2* was also at the top of our changed genes, along with related *Ccl1*, underscoring their sensitivity to skin perturbations, and tracing their expression to stem cells. However, in our system, when the entire immune repertoire was analyzed and quantified relative to bulge proximity, DCs were among the most consistently changed early immune respondents. Like M1 macrophages, DCs have surface receptors that respond to CCL1/2 (*Gombert et al., 2005*; *Moser et al., 2004*). Moreover, by in vitro co-culture and genetic targeting of the corresponding DC-CCL receptors, we showed that DCs use this crosstalk with HF stem cells to specifically congregate around the bulge. In this case, however, recruitment of DCs and (more variably) macrophages did not trigger a new hair cycle, pointing to a major difference between disrupting the barrier function of the BMP6-expressing inner niche layer and removing this layer altogether as in hair plucking which does initiate a new hair cycle.

During normal epithelial homeostasis, resident DCs can dialogue with Tregs (*Leventhal et al., 2016*; *Seneschal et al., 2012*), and thus may also be capable on their own to shape the adaptive

immune response that swarmed the damaged niche. Although our co-localization data and accumulation of Tregs with DCs around the bulge are supportive of this mechanism, our transcriptomics revealed that independently of immune cells, stem cells within a damaged niche are also well-equipped with a complex arsenal of cytokines and chemokines to communicate with both the adaptive and innate immune systems. Such chemokines include CCL1 (*Hoelzinger et al., 2010*; *Scharschmidt et al., 2017*) and CCL20, which are both potent stimuli for Tregs.

## Talking back: immune cell participation in containing the barrier breach

Recently, it was shown that in normal homeostasis, Tregs come into close proximity of the bulge towards the telogen to anagen transition, where in addition to BMP-inhibitors, WNTs and other activating cues, they help to fuel hair regeneration (*Ali et al., 2017*). Although Treg ablation studies suggested a role for Tregs in hair cycling, it has remained unknown whether such effects are direct or indirect. Our findings here showed that proliferative effects of Tregs on HF stem cells can be direct.

It is curious that in homeostasis (*Ali et al., 2017*) and hair plucking (*Chen et al., 2015*), stem cells are stimulated to proliferate and launch new hair growth, while in the face of a barrier breach, stem cells direct their efforts towards containing the damage. Based upon the evidence at hand, this difference likely resides in the plethora of growth factors, cytokines, reactive oxygen species, and lipid mediators that are generated by immune cells under different conditions (*Demehri et al., 2008*; *Yoon et al., 2016*).

Although the precise nature of this complex communication between the stem cells and immune cells is beyond the scope of the current study, we have shown that it is considerably more robust and diverse than that which functions in normal hair cycling or transient injury response. That said, the collective evidence points to an important role for stem cells in sensing and responding to their local environment. Their ability to subsequently recruit distinct immune cell repertoires enables stem cells to receive new sets of instructions that deviate from their normal homeostatic state, thereby channeling their efforts in new directions. This intricate feedback mechanism enables the recruited immune cells to repurpose stem cells from their normal job of making hair to instead participate in curbing the breached barrier.

This diversion of HF stem cells away from their normal regenerative function had two important outcomes. First, it added a new layer of AJ-competent cells to line/reinforce the damaged bulge. Second, because the activated stem cells transcribed an array of immune signaling genes, their amplification within the niche bolstered the local production of distress signals. Altogether, this captivating communication network appeared to be aimed at patching the breached barrier and limiting tissue damage that would otherwise be caused by chronic inflammation. Indeed, the ensuing immune cell-associated 'hyperthickening' phenotype was locally restricted to the site of the barrier breach (*Figure 8* and *Figure 6—figure supplement 2D*).

Since our breached HF barrier model was a genetic one, it was not surprising that the efforts of the HF stem cells to patch the barrier were futile, as evidenced by the bacterial infiltration seen in the *Cdh1*-null HFs and their surrounding tissue (*Figure 3*). Indeed, there are a number of genetic disorders involving breached skin barriers in which stem cells hyper-proliferate and yet are unable to repair the barrier. In this regard, it is intriguing to consider epidermolytic hyperkeratosis patients and mice, which harbor suprabasal epidermal keratin one and/or keratin 10 mutations. In this case, the healthy epidermal stem cells respond to the barrier breach, but they maintain a single layer of proliferative progenitors, while dramatically expanding the (defective) layers of terminally differentiating cells (*Fuchs et al., 1992*). By contrast, the proliferative *Cdh1* null HF stem cells cannot directly generate the terminally differentiated barrier cells (*Hsu et al., 2011*), and instead generate multiple layers of stem cells. While neither of these responses are successful, they likely reflect a general, evolutionarily conserved mechanism intrinsic within epithelial stem cells to respond to and repair barrier defects wherever they emerge.

## Interplay between stem cell adhesion, proliferation and immune response

Studies in *Drosophila* testis and ovary have long established a role for E-cadherin in adhering germline stem cells to their niche, a process critical to balancing stem cell self-renewal and differentiation

(*Song et al., 2002*; *Yamashita et al., 2003*). Although the roles of AJs in mammalian stem cell niches have not been studied extensively, the increased HF stem cell proliferation could reflect a direct response to the reduced AJ levels that appeared to emanate from the perturbed inner bulge: stem cell interface. Thus it was surprising that immune suppression kept the ectopic proliferation of telogen HF stem cells largely in check. These findings expose an additional layer of complexity in the proliferative control of HF stem cells not seen in the invertebrate stem cell systems studied to date.

Previous reports have described immune, hyperproliferative responses of mammalian epithelial tissues following genetic perturbations in various AJ components (*Perez-Moreno et al., 2006*; *Vasioukhin et al., 2001*). However, the responses documented occur directly at the level of genetic alterations and are thus largely cell-autonomous. The specific roles for α-catenin and p120-catenin in signaling also appear to be independent of barrier function. Notably, as we have shown here, the stem cells displaying a hyperproliferative response can themselves be wild-type and depend upon their border with damaged niche neighbors to trigger the subsequent cascade of events we identified.

Despite the critical role that the immune response plays to induce stem cell proliferation, the telogen-phase stem cells within the damaged niche still maintained in part a transcriptome reflective of cell proliferation even when the immune response was repressed. This aspect of their transcriptome paralleled that of their natural proliferative state in the hair cycle. However, in early anagen, the levels of hair cycle stimulatory factors outweigh the quiescence factors emanating from the bulge niche. By contrast, within a damaged telogen-phase niche, the quiescence factors still remained high, and in the absence of anagen-stimulating factors, the reduction in AJs and elevation in cell cycle transcriptome were not sufficient on their own to activate these stem cells. In this way, the proliferative trigger for repurposing the stem cells became contingent upon the unleashing of immune distress signals upon the failure of their niche neighbors to provide a proper barrier.

## Conclusions

Our studies exposed a fascinating interplay between the terminally differentiated cells that provide the niche barrier, and its neighboring stem cells that fuel regenerative processes in hair cycling and wound repair. Through their ability to respond to their damaged niche, stem cells are able to alter their program of gene expression in a way that both primes the stem cells for proliferation and enables them to recruit immune cells. In turn, the immune cells bestow new keratinocyte growth factors and cytokines on the local niche microenvironment. The strong presence of Tregs may also enable stem cell niches to limit tissue damage that might otherwise be caused by massive inflammation.

Our findings suggest that through their elevated cell cycle transcriptome, the primed stem cells within the damaged niche have a reduced threshold for proliferation and therefore selectively respond to recruited immune cells, even though normal homeostatic quiescent controls are still sufficient to keep other nearby cells (e.g. the hair germ) in check. This selective self-renewal behavior by stem cells sustains the immune response and also reinforces the damaged barrier with layers of cells whose intercellular junctions are intact. Our findings explain a number of pathological conditions involving perturbations in terminally differentiating cells that are manifested in healthy stem/progenitor cells.

## Materials and methods

### Mice and procedures

*Cdh3^{null}*, *Cdh1^{Flox}*, *Rosa26^{lox-STOP-lox-YFP}*, *Sox9-CreER*, *Krt15-CrePGR* and *Krt14-rtTA* were described previously (*Boussadia et al., 2002*; *Mao et al., 1999*; *Morris et al., 2004*; *Nguyen et al., 2006*; *Radice et al., 1997*; *Soeda et al., 2010*). *Sox9-CreER* was activated by tamoxifen (Sigma) administered either by intraperitoneal (i.p.) injection (75 µg/g body weight in corn oil) once a day for 3 days or by single topical application (1.5 mg in ethanol) on hair coat of mice. *Krt15-CrePGR* was activated by mifepristone (RU486, TCI America) administered by i.p. injection (75 µg/g body weight in sesame oil) and topically (12 mg in ethanol) on shaved dorsal back of mice once a day for 7 days. Antibiotics treatment was performed by feeding mice with chow containing 0.025% trimethoprim and 0.1242% sulfamethoxazole; feeding with antibiotics water containing metronidazole (500 mg/L, Sigma),

sulfamethoxazole (400 mg/L, Sigma), trimethoprim (100 mg/L, Sigma) and cephalexin (500 mg/L, Sigma) with Splenda sweetener to improve the taste of the water; and i.p. injection of vancomycin (0.75 mg, Sigma) and metronidazole (0.75 mg, Sigma) daily. *Sox9CreER* mice were treated with anti-biotics starting one week post-tamoxifen until time of harvest. Immunosuppression was achieved by daily i.p. injections of dexamethasone at 1 mg/kg body weight. rtTA was activated by feeding mice with doxycycline (2 mg/kg) chow. To track hair cycles (*Müller-Röver et al., 2001*), full-length telogen hairs were trimmed with electric clippers to reveal dorsal skin. HF entry into anagen was determined by darkening of skin and reappearance of hair. Completion of anagen and re-entry into telogen were determined by appearance of full-length hairs and loss of pigmentation in skin. EdU (5-ethynyl-2'-deoxyuridine; Thermo Fisher, 25 µg/g body weight) was injected i.p. into mice twice a day before obtaining skin biopsies or lethal administration of $CO_2$. All mice were maintained in a facility approved by The Association for Assessment and Accreditation of Laboratory Animal Care (AAA-LAC), and procedures were performed with protocols approved by Rockefeller University's institutional animal care and use committee (IACUC) members.

## Lentiviral constructs and injections of embryos

*Cdh1*-shRNA was obtained from the Broad Institute's Mission TRC-1/2 mouse library and cloned into LV-pLKO-PGK-H2B-RFP vector (*Moffat et al., 2006*). *Cdh1*-shRNA sequence is CCGAGAGAG TTACCCTACATA. For myc-tagged PCAD overexpression, *Cdh3* coding DNA sequence (CDS) containing myc tag sequence inserted after its pro-peptide sequence was synthesized and cloned into LV-TRE vector. To tag PCAD with myc at the C-terminal end, *Cdh3* CDS was amplified with myc inserted before its STOP codon, then cloned into LV-TRE vector. LV production, concentration and ultrasound-guided transduction of mouse embryos in utero were performed as described previously (*Beronja et al., 2010*).

## Histology, Immunofluorescence and Immunohistochemistry

To prepare sagittal skin sections for immunofluorescence microscopy, backskins were either freshly embedded and frozen in OCT, or pre-fixed in 4% paraformaldehyde (PFA) in PBS for 2 hr at 4°C then treated with 30% sucrose in PBS overnight at 4°C prior to embedding and freezing in OCT. Backskins were cryosectioned (12 µm) and sections from freshly embedded tissues were fixed for 10 min in 4% PFA. To prepare whole-mounts for immunofluorescence microscopy, subcutaneous fat was scraped from backskins, which were then incubated (dermis side down) on 2.5 U/ml dispase +20 mM EDTA for 2 hr at 37°C. Epidermis and HFs were separated from dermis, fixed in 4% PFA for 30 min at room temperature (RT) and permeabilized for 30 min in PBS + 0.5% Triton (PBST). Sections and whole-mounts were blocked for 1–2 hr at RT in 2% fish gelatin, 5% donkey serum, 1% BSA and 0.2% Triton in PBS. Primary antibody incubation was performed overnight at 4°C. Incubation with secondary antibodies conjugated to rhodamine red-X, Alexa 488, 546 or 647 was performed for 1–2 hr at RT. Mouse antibodies were incubated with M.O.M. block according to manufacturer's directions (Vector Laboratories). Images were acquired with Zeiss Axio Observer Z1 equipped with Apo-Tome.2 through a 20x air objective or Zeiss LSM780 laser-scanning confocal microscope through a 40x water objective.

To prepare sagittal skin sections for hematoxylin and eosin staining and immunohistochemistry, backskin was fixed in 4% PFA overnight at 4°C. Subsequent dehydration, paraffin embedding, sectioning and staining were performed by Histowiz Inc. Stained slides were scanned at 40x magnification using Aperio AT2 and visualized using Aperio Image Scope software.

Modified Gram staining method has been described by (*Becerra et al., 2016*). Briefly, paraffin sections were applied, in the following order, with crystal violet, Gram iodine, Gram decolorizer and safranin (Remel). To dehydrate, sections were immersed in 95% ethanol, 100% ethanol, then treated with alcoholic saffron (American Master Tech Scientific Inc.) before further dehydration with 100% ethanol and citrus clearing solvent. Images were acquired with an Axioplan two upright microscope equipped with a Spot Insight QE color digital camera through 40x N.A. 1.3 oil and 63x N.A. 1.4 oil objectives.

## Antibodies

The following antibodies and dilutions were used for immunofluorescence: ECAD (rabbit, 1:1000, Cell Signaling, AB_2291471), PCAD (goat, 1:400, Sigma, AB_355581), CD34 (rat, 1:100, eBioscience, AB_466426), K6 (guinea pig, 1:2000, Fuchs lab), TCF4 (rabbit, 1:250, Cell Signaling, AB_2199816), SOX9 (rabbit, 1:1000, Fuchs lab), LHX2 (rabbit, 1:1000, Fuchs lab), p120-catenin (mouse, 1:1000, Zymed, AB_87178), β-catenin (mouse, 1:1000, BD, AB_397555), α-catenin (rabbit, 1:1000, Sigma, AB_476830), DSG3 (mouse, 1:500, MBL, AB_591238), plakoglobin (mouse, 1:500, BD, AB_397649), DP1 and 2 (mouse, 1:500, Millipore, AB_93346), myc-tag (rabbit, 1:300, Cell Signaling, AB_490778), CLDN1 (rabbit, 1:500, Abcam, AB_301644), ZO1 (rabbit, 1:250, Zymed, AB_2533938), CD45 (rat, 1:100, Biolegend, AB_312969), MHC II (rat, 1:100, Biolegend, AB_493727), F4/80 (rat, 1:100, Biolegend, AB_893504), CD3 (rat, 1:100, Biolegend, AB_1877072), FOXP3 (rat, 1:25, eBioscience, AB_467576), GFP (chicken, 1:2000, Abcam, AB_300798), pSMAD1 (rabbit, 1:100, Cell Signaling, AB_331671) and phospho-p65 (rabbit, 1:50, Cell Signaling, AB_330559). Phalloidin (1:100, Thermo Fisher) was used to detect F-actin. Nuclei were stained with 4'6'-diamidino-2-phenylindole (DAPI). EdU click-iT reaction was performed according to manufacturer's directions (Thermo Fisher). β-catenin antibody (rabbit, 1:50, Cell Signaling, AB_11127855) was used for immunohistochemistry.

## Flow cytometry

To prepare single cell suspensions from telogen backskin, subcutaneous fat was scraped off with a scalpel and backskin was placed (dermis side down) on 0.25% trypsin (Gibco) for 35–45 min at 37°C. If inner bulge cells were to be isolated, backskin was placed on trypsin overnight at 4°C, then 1 hr at 37°C. To prepare single cell suspensions from anagen backskin, backskin was placed (dermis side down) on collagenase (Sigma) for 45 min at 37°C, dermal side was scraped off with a scalpel, and remaining epidermal side was transferred to trypsin for 20 min at 37°C. For both telogen and anagen backskins, trypsinized HF and epidermal cells were then scraped off gently and filtered with strainers (70 μm, followed by 40 μm).

Dissociated cells were incubated with antibodies for 20 min at 4°C. The following antibodies were used: CD34-eFluor660 (1:100, eBioscience, AB_10596826), α6-PE (1:100, BD Bioscience, AB_396079), α6-PerCP-Cy5.5 (1:250, Biolegend, AB_2249260), β1-PE-Cy7 (1:400, eBioscience, AB_1234962), Sca1-PerCP-Cy5.5 (1:1000, eBioscience, AB_914372), Sca1-APC-Cy7 (1:1000, Biolegend, AB_10645327), CD200-PE (1:300, eBioscience, AB_1907362), CD45-biotin (1:200, eBioscience, AB_466446), CD31-biotin (1:200, eBioscience, AB_466423), CD117-biotin (1:200, eBioscience, AB_466569), CD140a-biotin (1:200, eBioscience, AB_466606) and Streptavidin-PE-Cy7 (1:500, eBioscience, AB_1011648). DAPI was used to exclude dead cells. Cell purification was performed on FACS Aria sorters equipped with Diva software (BD Bioscience).

To analyze immune cells, backskin was minced in RPMI1640 media with L-glutamine (Gibco), 1 mM sodium pyruvate (Lonza), 10 mM acid-free HEPES (Gibco), 100 U/ml penicillin and streptomycin (Gibco). Liberase TL (Roche) was added to the media (25 g/ml) and backskin was digested for 120 min at 37°C. Digestion was stopped by addition of 20 ml of 0.5 M EDTA (Life Technologies) and 1 ml of 10% DNase (Sigma) solution. Dissociated cells were filtered with 70 μm strainer and stained with the following antibodies: Ly6c-FITC (1:100, Biolegend, AB_1186134), Ly6g-PE (1:200, Biolegend, AB_1186104), CD11c-PE-Cy7 (1:150, Biolegend, AB_493569), CD11b-PacBlue (1:300, Biolegend, AB_755985), MHCII-AF700 (1:300, Biolegend, AB_493727), CD45-AF750 (1:100, Biolegend, AB_2572115), CD64-PerCP-Cy5 (1:200, Biolegend, AB_2561962), TCRβ-PerCP-Cy5.5 (1:200, Biolegend, AB_1575176), γδTCR-APC (1:400, Biolegend, AB_1731813) and MerTK-PE (1:100, Biolegend, AB_2617035). Dead cells were excluded using LIVE/DEAD Fixable Blue Dead Cell Stain Kit (Molecular Probes) for UV excitation. FACS analyses were performed using LSRII FACS Analyzers and results were analyzed using FlowJo software.

## ELISA

FACS-purified cell protein lysates were prepared using RIPA buffer containing protease inhibitor (Roche) and split equally for ECAD-ELISA and PCAD-ELISA. ECAD-ELISA was performed as per manufacturer's protocol (R and D Systems). PCAD-ELISA was performed as per ECAD-ELISA protocol, but instead using PCAD monoclonal antibody (rat, Life Technologies, AB_2533006) as capture antibody, PCAD-biotin polyclonal antibody (goat, R and D Systems, AB_442232) as detection antibody,

and recombinant mouse PCAD Fc chimera protein (R and D Systems) to calibrate standard curve for quantification.

## Immunoblotting

Gel electrophoresis of FACS-purified cell protein lysates was run on 4–12% NuPAGE Bis-Tris gradient gel (Thermo Fisher) and transferred to nitrocellulose membrane (Amersham). Membrane was blocked in TBS containing 2% bovine serum albumin (BSA) and 0.1% Tween-20 for 1 hr at RT, incubated with primary antibodies overnight at 4°C and with secondary antibodies conjugated to horseradish peroxidase (HRP) for 1 hr at RT. HRP activity was detected using enhanced chemiluminescence (ECL) substrate (Amersham). Quantification of bands was performed on ImageJ software. The following antibodies were used: ECAD (rabbit, 1:1000, Cell Signaling, AB_2291471), PCAD (goat, 1:1000, R and D Systems, AB_355581), p120 (mouse, 1:1000, Zymed, AB_87165), β-catenin (mouse, 1:1000, BD Biosciences, AB_11127855), α-catenin (rabbit, 1:4000, Sigma, AB_476881), claudin 1 (rabbit, 1:500, Abcam, AB_301644), ZO1 (rabbit, 1:500, Life Technologies, AB_2533938) and GAPDH (mouse, 1:4000, Abcam. AB_2107448).

## 16S FISH and Tissue-Clearing

To prepare samples for 16S-FISH, backskins, including those of germ-free mice as negative control, were fixed in 4% PFA overnight at 4°C, then washed with PBS before being permeabilized in 0.3% PBST overnight at 4°C. Backskins were additionally fixed and permeabilized through a methanol series: 100% for 4 × 5 min, 100% for 1 × 30 min, 75% for 1 × 5 min, 50% for 1 × 5 min, and 30% for 1 × 5 min, followed by 3 × 5 min of 0.3% PBST washes. To probe for bacterial 16S, FISH with hybridization chain reactions (HCR) was employed (*Choi et al., 2010*). Samples were pre-hybridized with probe hybridization buffer for 30 min at 45°C before incubating with 1 pmol of each probe, mixed together, overnight at 45°C. The sequences of the probes (probeBase) were:

EB338: GCUGCCUCCCGUAGGAGU
Univ1390: GACGGGCGGUGUGUACAA
Scrambled (non-EUB, negative control): ACUCCUACGGGAGGCAGC

Following hybridization, samples were washed with probe wash buffer for 2 × 5 min and 2 × 30 min at 45°C, then washed with 5X SSCT (containing 0.1% Tween-20) for 3 × 5 min at RT. Samples were pre-amplified with amplification buffer for 30 min at RT before incubating with 30 pmol of fluorescently labeled hairpins overnight at RT. Samples were washed with 5X SSCT for 2 × 5 min, 2 × 30 min and 1 × 5 min at RT before fixation in 4% PFA for 5–15 min at RT and incubation with DAPI (2 µg/ml) overnight at 4°C. Samples were then washed with 0.3% PBST before being cleared for imaging. Probe hybridization buffer, probe wash buffer and probes were purchased from Molecular Technologies.

For tissue clearing, backskins were transferred through increasing concentrations of ethanol diluted in distilled water and adjusted to pH9.0: 30% for 20 min, 50% for 20 min, and 70% for 20 min. Backskins were then incubated in 100% ethanol for 1 hr before transferring into ethyl cinnamate (Sigma) for clearing.

Images were acquired with an inverted spinning-disk confocal system driven by iQ Live Cell Imaging software (Andor) using a 20x N.A. 0.75 air objective and 561 nm, 642 nm and UV laser lines, and analyzed using Imaris software. For spot analysis and quantification, sphere sizes were set to 2.5–3, and the estimated x-y diameter for spots was 0.6–1.5 µm.

## RNA purification, RNA-seq and qRT-PCR

WT bulge RNA-seq data is from *Lay et al. (2016)*. For all others, total RNA was extracted from FACS-purified cells by directly sorting cells into Trizol$^{LS}$ (Sigma) and processing with Direct-Zol RNA mini-prep kit (Zymo Research). RNA quality was determined using an Agilent 2100 Bioanalyzer and all samples sequenced had RNA integrity numbers > 8. mRNA library preparation using Illumina TrueSeq mRNA sample preparation kit and single-end sequencing on Illumina HiSeq 4000 were performed at Weill Cornell Medical College Genomic Core Facility (New York). Alignment of reads was done using STAR version 2.5.2a (Spliced Transcripts Alignment to a Reference) (*Dobin et al., 2013*) and transcripts were annotated using Gencode release M10 of the mouse genome. Differential gene expression analysis was performed with DESeq2 package version 1.18.0 (*Love et al., 2014*) using

the gene count output from STAR read aligner. Gene ontology analysis was performed using DAVID (*Huang et al., 2009a*; *Huang et al., 2009b*).

To perform qRT-PCR, equal amounts of RNA were reverse transcribed using Superscript III with oligo-dT primer (Invitrogen) or Superscript VILO (Invitrogen) cDNA Synthesis kits. cDNAs were mixed with indicated primers and SYBR green PCR Master Mix (Sigma), and qRT-PCR was performed on an Applied Biosystems 7900HT Fast Real-Time PCR system. cDNAs were normalized to equal amounts using primers against *Ppib*. The following primer sequences were used (5′ → 3′):

*Ccl1* forward: AGTGTTACAGAAAGATGGGCTC, *Ccl1* reverse: GAGGACTGAGGGAAACTGC
*Ccl2* forward: GTCCCTGTCATGCTTCTGG, *Ccl2* reverse: GTGATCCTCTTGTAGCTCTCC
*Ppib* forward: GTGAGCGCTTCCCAGATGAGA, *Ppib* reverse: TGCCGGAGTCGACAATGATG
*Bmp6* forward: GGGGCTCCGGTTCTTCAGA, *Bmp6* reverse: GGACGTACTCGGGGATTCATAAGGT,
*Fgf18* forward: CTGTGCTTCCAGGTTCAGGT, *Fgf18* reverse: TGCTTCCGACTCACATCATC,

## Lucifer yellow assay to assess barrier function

Skin biopsies were treated with dispase with EDTA, and epidermis was separated from dermis, in the same way as preparing tissue for whole-mount immunofluorescence. HFs in the epidermal fraction were then submerged in 1 mM Lucifer yellow (Thermo Fisher) in PBS for 3 hr at 37°C before fixation, mounting and imaging.

## Cell culture studies

HF stem cells were FACS purified and co-cultured with feeder fibroblasts in 300 μM Ca$^{2+}$ as described previously (*Nowak and Fuchs, 2009*). For growth curve assays, equal numbers of *Cdh1* cKO and control HF stem cells were cultured in triplicates in the absence or presence of 10 nM dexamethasone, and cell numbers were counted every other day from Days 5 to 9.

For transwell migration assay, bone marrow-derived dendritic cells (BMDCs) or bone marrow-derived macrophages (BMDM) were obtained from femur and tibiae bone marrow of 8-week-old C57BL/6 mice and cultured in BMDC/macrophage medium (RPMI 1640 with L-glutamine, 10% heat-inactivated FBS, 10 mM HEPES, and 100 U/ml penicillin/streptomycin) at a density of $1 \times 10^6$ cells/ml with addition of 20 ng/ml recombinant murine GM-CSF (R and D Systems) and 10 ng/mL recombinant murine IL-4 (R and D Systems) for BMDC and 10 ng/ml recombinant murine M-CSF (R and D Systems) for BMDM derivation. The maturation of BMDCs or BMDM was induced by 100 ng/ml of LPS 24 hr prior to co-culture with HF stem cell medium. $5 \times 10^5$ mature WT or *CCR2*-nullBMDCs or BMDM were seeded onto upper chambers of 5 μm pore size in transwell plates with or without isotype control antibody or CCR8 blocking antibody (Biolegend, AB_2566246). WT or *Cdh1*-null HF stem cells were cultured in lower chambers and incubated for 3.5 hr at 37°C. BMDCs or BMDM that migrated into the bottom chamber were harvested and counted by flow cytometry, using CD45-APC (1:500, eBioscience, AB_469392) and DAPI to exclude dead cells.

For co-culture proliferation assays, BMDCs were isolated in the same way as described above for transwell migration assay. Naive CD4+ T cells were isolated from the spleen of the 8-week-old C57BL/6 mice by Mojo CD4 T cell Negative Selection Kit (Biolegend), followed by in vitro activation on plates coated with 10 μg/ml anti-CD3 antibody (Thermo Fisher Scientific, AB_468847) and 2 μg/ml anti-CD28 antibody (Thermo Fisher Scientific, AB_468921), with or without 5 ng/ml TGF-β1 (R and D Systems), and incubation for 5 days to obtain regulatory T cells (Treg) and conventional T cells (Tconv) respectively. Tregs were then enriched by FACS using CD45-APC/Cy7 (1:200, Biolegend, AB_312981), CD4-PE/Cy7 (1:200, Biolegend, AB_312707), CD25-APC (1:200, Biolegend, AB_2280288), CD127-PE (1:200, Biolegend, AB_493509), and DAPI to exclude dead cells. Both Tconv and FACS-purified Tregs were cultured in T cell medium (RPMI-1640 with L-glutamine, 10% heat-inactivated FBS, 1 mM HEPES, 1 mM sodium pyruvate, 100 U/ml penicillin/streptomycin, and 55 μM 2-mercaptoethanol). To measure the effects of BMDCs, Tconv and Tregs on HF stem cell proliferation, 5000 *Cdh1* cKO HF stem cells were seeded onto feeder fibroblast layers in lower chambers of transwell plates. 2 days later, HF stem cells were co-cultured with $2 \times 10^5$ LPS-treated BMDCs seeded in BMDC media, or $1 \times 10^4$ Tconv or FACS-isolated Tregs seeded in T cell media in upper chambers of 0.8 μm pore size for 4–5 days before their numbers were counted.

## Electron microscopy

For EM, backskins were fixed in 2% glutaraldehyde, 4% PFA, and 2 mM $CaCl_2$ in 0.05 M sodium cacodylate buffer, pH7.2, at RT for >1 hr, post-fixed in 1% osmium tetroxide, and processed for Epon embedding. Ultrathin sections (60–65 nm) were counterstained with uranyl acetate and lead citrate. EM images were taken with a transmission electron microscope (Tecnai G2-12; FEI) equipped with a digital camera (Ultrascan, Gatan Inc.).

## Statistics and data availability

Data were analyzed and statistics were performed using unpaired two-tailed Student's t-test (when comparing two groups), one-way ANOVA with Tukey's post-hoc test (when comparing >2 groups) or two-way ANOVA with Sidak test (for quantifications of EdU +HF stem cell numbers) in GraphPad Prism versions 6 and 7, with 0.05 level of confidence being accepted as a significant difference. The accession number for the RNA-seq data reported in this paper is NCBI GEO: GSE106767.

## Acknowledgements

We thank Fuchs' lab colleagues F Garcia Quiroz for input on PCAD overexpression construct design and barrier assays; N Gomez for bioinformatics expertise; I Matos, V F Fiore, S Baksh and M Tierney for discussions; A Sendoel, S Ellis, M Laurin and F Garcia Quiroz for critical reading of the manuscript; J Levorse for *in utero* lentiviral injections; L Polak, M Sribour and L Hidalgo for mouse husbandry and procedures; E Wong and S Mereby for genotyping assistance. We thank D Mucida (Rockefeller) for germ-free mice and H Choi (Caltech) for 16S probes and reagents. We thank Rockefeller's facilities: Comparative Bioscience Center for care of mice in accordance with National Institutes of Health (NIH) guidelines; Flow Cytometry Resource Center for FACS; Bioimaging Resource Center for confocal and light microscopy. We also thank Weill Cornell Medical College Genomics Resources Core Facility for performing RNA-seq. EF is an investigator of the Howard Hughes Medical Institute. KL is a Fellow of the Singapore Agency for Science, Technology and Research (A*STAR). SY and SBL are NIH Ruth L Kirschstein Predoctoral Fellows. SG-C is a fellow of the Human Frontier Science Program. YM is a Jane Coffin Child Fellow. SN is a Dale Frye Damon Runyon Cancer Research Foundation Fellow (DRG-2183–14). This study was supported by NIH grants R01-AR27883 and R01-AR050452 (EF) and L'Oreal USA For Women in Science (SN).

## Additional information

### Competing interests

Elaine Fuchs: Reviewing editor, *eLife*. The other authors declare that no competing interests exist.

### Funding

| Funder | Grant reference number | Author |
| --- | --- | --- |
| National Institutes of Health | R01-AR27883, R01-AR050452 | Elaine Fuchs |
| L'Oreal USA | | Shruti Naik |

The funders had no role in study design, data collection and interpretation, or the decision to submit the work for publication.

### Author contributions

Kenneth Lay, Conceptualization, Data curation, Formal analysis, Supervision, Validation, Investigation, Visualization, Methodology, Writing—original draft, Project administration, Writing—review and editing, Conceptualized the study, Designed the experiments, Wrote the manuscript, Performed Cdh1 cKO, Cdh1 knockdown, Cdh3-null and Cdh3 over-expression studies, FACS, immunoblotting, and RNA-seq, Performed cell culture studies, Analyzed immune cell infiltration and significance, Carried out histological analysis and qRT-PCR; Shaopeng Yuan, Data curation, Formal analysis, Validation, Investigation, Visualization, Methodology, Project administration, Performed cell culture

studies, Analyzed immune cell infiltration and significance, Carried out histological analysis and qRT-PCR, Assisted TH with performing the ELISAs; Shiri Gur-Cohen, Data curation, Formal analysis, Validation, Investigation, Visualization, Methodology, Performed 16S-FISH, Analyzed immune cell infiltration and significance; Yuxuan Miao, Data curation, Formal analysis, Validation, Investigation, Visualization, Methodology, Performed cell culture studies, Analyzed immune cell infiltration and significance; Tianxiao Han, Validation, Investigation, Carried out histological analysis and qRT-PCR, Performed the ELISAs with SY's assistance, Helped with Cdh3 over-expression study; Shruti Naik, Formal analysis, Funding acquisition, Validation, Investigation, Visualization, Methodology, Analyzed immune cell infiltration and significance; H Amalia Pasolli, Formal analysis, Validation, Investigation, Visualization, Methodology, Analyzed immune cell infiltration and significance, Performed electron microscopy and analyzed the results; Samantha B Larsen, Investigation, Analyzed immune cell infiltration and significance; Elaine Fuchs, Conceptualization, Resources, Data curation, Software, Formal analysis, Supervision, Funding acquisition, Methodology, Writing—original draft, Project administration, Writing—review and editing, Conceptualized the study, Designed the experiments, Wrote the manuscript

## Author ORCIDs
Kenneth Lay http://orcid.org/0000-0001-8843-5567
Shaopeng Yuan https://orcid.org/0000-0001-9554-1325
Elaine Fuchs http://orcid.org/0000-0002-0978-5137

## Ethics
Animal experimentation: All mice were maintained in a facility approved by The Association for Assessment and Accreditation of Laboratory Animal Care (AAALAC), and procedures were performed with protocols approved by Rockefeller University's institutional animal care and use committee (IACUC) members.

## Decision letter and Author response
Decision letter https://doi.org/10.7554/eLife.41661.025
Author response https://doi.org/10.7554/eLife.41661.026

## Additional files

### Supplementary files
• Transparent reporting form
DOI: https://doi.org/10.7554/eLife.41661.019

### Data availability
RNA-sequencing data have been deposited in GEO under accession number GSE106767

The following dataset was generated:

| Author(s) | Year | Dataset title | Dataset URL | Database and Identifier |
|---|---|---|---|---|
| Lay K, Yuan S, Gur-Cohen S, Miao Y, Han T, Naik S, Pasolli HA, Larsen SB, Elaine Fuchs E | 2018 | RNA-sequencing data from Stem cells repurpose proliferation to contain a breach in their niche barrier | https://www.ncbi.nlm.nih.gov/geo/query/acc.cgi?acc=GSE106767 | Gene Expression Omnibus, GSE106767 |

The following previously published dataset was used:

| Author(s) | Year | Dataset title | Dataset URL | Database and Identifier |
|---|---|---|---|---|
| Lay K, Kume T, Fuchs E | 2016 | RNA-seq analysis of hair follicle stem cell transcriptome upon loss of the transcription factor FOXC1 | https://www.ncbi.nlm.nih.gov/geo/query/acc.cgi?acc=GSE77256 | NCBI Gene Expression Omnibus, GSE77256 |

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
