## [Decision Letter]

Thank you for sending your article entitled "Stem cells repurpose proliferation to contain a breach in their niche barrier" for peer review at *eLife*. Your article has been evaluated by three peer reviewers, one of whom is a member of our Board of Reviewing Editors, and the evaluation has been overseen by Marianne Bronner as the Senior Editor.

As you can see below, the reviewers had several concerns including the link between the loss of E-cadherin and the immune system, the mechanisms by which CCL ligands are elevated and function, and whether other transcriptional pathways are altered in the E-cadherin cKO mice.

*Reviewer #1:*

This manuscript addresses the role of cellular junctions on stem cell dynamics using the hair follicle (HF) as a model system. The authors show that disruptions in junctions through genetic loss of E-cadherin in hair follicle stem cells leads to activation of immune cells and proliferation. Interestingly, the stem cells do not differentiate but remain in the niche to reinstate the barrier. The manuscript contains several interesting points but the experiments that point to a role of the immune system are not specific and will likely require additional experiments that are beyond the shortened revision time for *eLife*.

1) Figure 4. The graph of Tregs is not labelled on the figure or the legend. What is the difference between the open circles and the triangles?

2) Figure 5. Which cKO mouse is used for the expression analysis? The legend and/or figure should reflect the genetic model used for this experiment.

3) The difference between macrophage vs. DC cell numbers in the cKO is confusing. Since CCR2 recruits macrophages and DCs, the upregulation of these genes should elevate macrophages in the cKO mice. These seemingly disparate results should be clarified in the manuscript.

4) The experiments using dexamethasone (Dex) are quite non-specific for inhibiting immune cells. It is known that Dex can alter DP cells (Kwack et al., 2017) and other cell types in the skin. The authors do show that direct interaction of Tregs and bulge cells induces their proliferation, but this was shown recently (Ali et al., 2017) using careful genetic models of cellular depletion. The authors should perform specific experiments that implicate the immune cells and/or specific immune cells.

5) The authors do not perform functional analysis of BMP/FGF signaling in proliferation in their model.

6) Does the proliferation that is induced in the bulge regenerate the barrier function in the cKO mice?

7) In the figure legends the authors mention n=4 mice. Is that 4 mice per genotype or 2 mice per genotype?

*Reviewer #2:*

In the manuscript Lay et al. discover an important interplay between HFSC and immune cells following breaches in the inner bulge niche. Using several strategies, they demonstrate that E-cadherin is important for maintaining adherens junctions, tight junctions, and integrity of the niche, but does not affect niche cell numbers in the bulge. Ablation of *Cdh1* specifically in bulge stem cells did not cause the stem cell proliferation phenotype indicating that niche signaling is responsible for stem cell proliferation and expansion of bulge layers in *Cdh1* KO mice. Treating *Cdh1* KO mice with various antibiotics did not prevent stem cell proliferation, demonstrating that bacterial infiltration is not responsible for stem cell proliferation following niche breach. Dendritic cells (DCs) and αβ T cells, but not macrophages or γδTCR, showed marked and significant recruitment to breached stem cell niches. Immune cell recruitment was mediated by an upregulation of chemokines in *Cdh1* KO HFSC and WT HFSC near *Cdh1* KO inner bulge niche, revealed with RNA-Seq using FACS sorted populations and validated with in vitro migration assays. *Ccl1* and *Ccl2*, established chemokines for DCs, were the top two most highly upregulated genes in the stem cell cohort. Increases in *Ccl1* and *Ccl2* occurred in *Cdh1* KO HFSC in the presence of antibiotics, demonstrating again that the breached niche response is independent of bacterial infiltration. Furthermore, DC recruitment by HFSC is dependent on DC CCR2 expression, the receptor to *Ccl2*. Lastly, immunosuppression prevented the HF hyper thickened bulge phenotype and HFSC proliferation, revealing a role for immune cell infiltration in breached niche protection independent of bacterial infiltration. The experiments are controlled well, observations validated using several strategies, and conclusions strongly supported by the data. The manuscript will be of interest to a general audience and is highly novel. I have a few remaining questions.

1) Does *Ccl1* and *Ccl2* expression increase in WT HSFC surrounding *Cdh1* KO inner bulge in antibiotic conditions?

2) Why is an increase in HFSC *Ccl1* important? *Ccl1* interacts with CCR8, expressed by DCs, T cells and monocytes. Were in vitro migration assays using CCR8 KO DCs or T cells performed? Does *Ccl1* have a role in breached niche protection?

*Reviewer #3:*

In this manuscript, Lay and colleagues studied the effect of E-cadherin loss in hair follicle stem cells and their niche. Using sophisticated mouse models to delete E-cadherin in the stem cells and niche either collectively or separately, they have made fascinating observations that hair follicle stem cells can sense the barrier breach and engage in a hyper-proliferative state without triggering hair follicle growth. They have further delineated the role of bacterial infiltration and immune cell accumulation followed by the barrier breach and demonstrated that only immune cell accumulation but not bacterial infiltration leads to abnormal stem cell division. The data are of very high quality and convincing.

---

## [Author Response]

As you can see below, the reviewers had several concerns including the link between the loss of E-cadherin and the immune system, the mechanisms by which CCL ligands are elevated and function, and whether other transcriptional pathways are altered in the E-cadherin cKO mice.Reviewer #1:This manuscript addresses the role of cellular junctions on stem cell dynamics using the hair follicle (HF) as a model system. The authors show that disruptions in junctions through genetic loss of E-cadherin in hair follicle stem cells leads to activation of immune cells and proliferation. Interestingly, the stem cells do not differentiate but remain in the niche to reinstate the barrier. The manuscript contains several interesting points but the experiments that point to a role of the immune system are not specific and will likely require additional experiments that are beyond the shortened revision time for eLife.

We appreciate the reviewer’s comments and also recognized in reading some of their remarks that there were experiments/data presented that needed clarification. In fact, our studies underscoring a role for the immune system *are* specific for the particular question we try to address, as we explain in response to several of the reviewer’s concerns raised below.

1) Figure 4. The graph of Tregs is not labelled on the figure or the legend. What is the difference between the open circles and the triangles?

We thank the reviewer for pointing out this inadvertent omission. The open-circle dataset included an outlier that was consistent with our findings (an increase in Tregs upon barrier breach) but made the difference statistically insignificant. The triangle dataset excluded the outlier, in which case the difference between cKO and control was statistically significant. Since the outlier is part of the experimental data, and yet our conclusions are clear, we felt this was the fairest way to present these data. We have now indicated this in the legend for Figure 4D-F.

2) Figure 5. Which cKO mouse is used for the expression analysis? The legend and/or figure should reflect the genetic model used for this experiment.

We thank the reviewer for pointing out this poorly explained point, which we now clarify in the legend/Materials and methods. In all cases, our “controls” are always matched littermate heterozygous animals. For Figure 5A, we use low dose Tam and *SOX9CreER* during mid-telogen to analyze the difference between the inner bulge cells lacking *Cdh1* and their heterozygous WT counterparts. For Figure 5B and C, we use high dose Tam and *SOX9CreER* during mid-telogen and determine the HFSC signature in HFs whose bulge is either WT (heterozygous) or mutant for *Cdh1*. For Figure 5D, top Venn Diagram, we use low dose tamoxifen with SOX9CreER and compare how WT HFSCs differ in gene expression when they experience a WT versus a mutant niche. Principle component analyses in Figure 5E reveal that the signature is very different from simply WT HFSCs during their natural phase of proliferation (Anagen II/III). For Venn Diagram in Figure 5D (bottom), we show that the inner bulge and HFSCs respond very differently to the breached barrier.

In Figure 5F, we are taking the top genes from the inner bulge “immune crosstalk” signature (cKO inner bulge signature vs. control inner bulge) and then comparing their expression levels to the HFSC expression levels. From these data, it is clear that the levels of even these genes are higher in the HFSCs than the inner bulge niche.

In Figure 5G, we take the top genes from the HFSC “immune crosstalk” signature (*Cdh1*-null HFSCs vs. control HFSCs) and then compare their levels to those of the inner bulge whose *Cdh1* gene is ablated. Again, the gene expression levels are higher, substantially so, over the inner bulge expression levels. In Figure 5G, for comparative purposes, we also add an analysis of how these top HFSC “immune crosstalk” genes differ when the HFSCs are either *Cdh1* null (red bars) or WT (green bars).

The importance of performing the experiments in this way was to maintain the same genetic background throughout and analyze every possible combination. From the collective analyses, we could clearly demonstrate that the “immune crosstalk” signature was seen irrespective of whether the HFSCs were WT or *Cdh1* null, but that it was in all cases dependent upon the status of the inner bulge niche (either breached or WT).

We have now revised the Figure 5 legend to indicate the mouse models used for the various RNA-seq analyses.

3) The difference between macrophage vs. DC cell numbers in the cKO is confusing. Since CCR2 recruits macrophages and DCs, the upregulation of these genes should elevate macrophages in the cKO mice. These seemingly disparate results should be clarified in the manuscript.

I presume the reviewer meant to say “...since CCL2 recruits macrophages and DCs...”. Since we discovered CCL2 specifically expressed by HFSCs within a breached barrier niche, indeed, we could have expected to see either an increase in macrophages or DCs or both.

In contrast to many studies that have been performed on the skin (such as the Cheng-Ming Chuong study reporting elevated CCL2 (location unknown) and macrophages in skin following hair plucking, which induces mechanical damage and removes the BMP6/FGF18-expressing niche cells), we performed a complete immune cell analyses, and carefully quantified by FACS to identify which of the immune cell types infiltrate the skin soon after a breach of the HFSC niche. We also then used immunofluorescence to localize which of these immune cells were specifically localized to the HFSCs, which in our case was the direct source of CCL2. In so doing, it was clear that the main immune cell differences in our breached barrier model were in DCs and Tregs.

Although macrophages and DCs share many markers in common, we define DCs by CD45+CD11b+ CD64-Mertk-Ly6C-Ly6G-MHCII+CD11c+ cells, and CD45+CD11b+MHCII+CD64+MertK+ cells as Macrophages. We have now added these distinguishing markers to our FACS plot analyses so that readers can see the full regalia of our analyses. There are several studies that report that cytokine milieu can skew the populations of Macs vs. DCs, and we cite these papers, as they have relevance to our study. Additionally, we added a sentence to make sure that we don’t give the reader the misimpression that macrophages are irrelevant, since they are certainly present in normal skin, and we did see some signs of macrophage increases in the breached niche (data in Figure 4—figure supplement 1E and F). It is just that in our experiments, the levels of macrophage increases were quite variable and analyses indicated no significant differences (see Figure 5D)(note that we did analyze multiple mice – 6 by FACS and more by immunofluorescence – and this difference remained variable, whereas DC increases were significant). We’ve now expanded our FACS profiles in Figure 4—figure supplement 1E and adjusted the text (subsection “Immune cells are recruited to the breached niche independently of bacteria”, fourth paragraph) to point out that macrophages were seen as we show in Figure 4—figure supplement 1 immunofluorescence images, but the numbers were not sufficiently consistent to be statistically significant.

4) The experiments using dexamethasone (Dex) are quite non-specific for inhibiting immune cells. It is known that Dex can alter DP cells (Kwack et al., 2017) and other cell types in the skin. The authors do show that direct interaction of Tregs and bulge cells induces their proliferation, but this was shown recently (Ali et al., 2017) using careful genetic models of cellular depletion. The authors should perform specific experiments that implicate the immune cells and/or specific immune cells.

We thank the reviewer for their comments.

Indeed, we did carefully weigh the options to use dexamethasone versus genetic targeting of specific immune cell populations. The rationale we chose will be clearer having better clarified the text. Here are the two questions we asked and addressed with dexamethasone:

1) Do the infiltrating immune cells cause the transcriptome changes that we see in stem cells that experience a breached niche or is this rooted in the breached niche? Total immune suppression was needed to address this question, and we show that the aberrant cell cycle and inflammatory transcriptome of the telogen-phase stem cells occurs independently of the immune cell infiltration (Figure 6E-G). Since total immune ablation did not impact the HFSC transcriptome, specific immune ablation experiments were not relevant in addressing this point.

2) Is the proliferation that we see in telogen phase stem cells that experience a barrier breach due to the immune infiltration or the barrier breach? Total immune suppression was necessary to demonstrate this and we show that the immune cell infiltration was necessary (Figure 6B).

The reviewer raises an important point, namely that dexamethasone not only suppresses the immune system, but can affect other cell types. Regarding the Experimental Dermatology paper that the reviewer refers to, and which picks up on an earlier Paus paper, it has been reported that Dex can affect full anagen-phase mouse follicles, where it can lead to early entry into catagen. In our case, however, we focus on mid-telogen, where no adverse effects of Dex have been previously described, and critically for our study here, we do not see overt changes in the effects of Dex on our control WT mice. Of equal importance, we always use matched littermate controls in our experiments, and our focus is on differences that are exclusive to the skin whose niche barrier has been breached.

In this context, since both immune infiltration and aberrant telogen phase proliferation were eliminated by Dex, it was important to show that Dex alone does not impair HFSC proliferation. We did this experiment by culturing HFSCs under conditions where they proliferate and demonstrating that Dex does not affect proliferation status of HFSCs (Figure 6B). We explain this rationale more clearly in the text, and have added the Experimental Dermatology reference suggested by this reviewer.

We also weighed the options to use genetic targeting of specific immune cell populations versus coculturing of specific types of immune cells with HF stem cells. The reviewer highlights the Ali Cell paper, demonstrating that resident Tregs are present in normal skin and that they have a positive effect on normal anagen. However, our question was whether there is direct crosstalk between Tregs and stem cells and/or between DCs and stem cells, and since Tregs are known to impact other cell types, including other immune cells and since they could be mediating their effects indirectly through action on other skin populations, we felt this strategy may not be optimal. Rather, to address whether the immune component is directly involved in cross-talk with stem cells, it is essential to examine the effects of specific immune cells on HFSCs and the effects of the HFSCs on the specific immune cells in isolation. In this case, because DCs and Tregs are the major immune cells that swarm the breached niche, we purified and prepared DCs and Tregs, and then performed co-culture assays with HFSCs, purified from the breached niche. The outcome of these data are clear: DCs can be recruited by HFSCs in a transwell migration assay, but only if they have the CCR2 receptor to sense CCL2 (Figure 5I). By contrast, Tregs affect HFSC proliferation (Figure 6D). While our findings on Tregs are consistent with those of Ali et al., our experiments show for the first time that the effects are direct (conditionally knocking out Notch ligand in Tregs in vivo similarly did not address whether the effect the authors saw was directly between Tregs and HFSCs).

Since the co-culture experiments were definitive, and since they were the best way to achieve our goal here, i.e. to demonstrate direct cross-talk between HFSCs and immune cells, we did not pursue immune cell ablation studies. However, it is important to consider that the proposed experiments cannot simply be added as supplementary data providing ‘supportive evidence’ – they are neither trivial nor cheap. We already have 4 genetically targeted alleles (Sox9CreER, R26YFP, *Cdh1*fl/fl) and DCs would add at least another two alleles, while Tregs would add another allele. We do discuss the Ali et al. Cell paper and emphasize better in the text the virtues of the co-culture experiments versus the immune ablation experiments. As a final note, we also make certain to point out that, in skin whose HF barrier is breached, there is an even more robust and also highly niche-focused elevation in Tregs that is not seen in normal skin.

We have now revised the Results section entitled “Stem cell proliferation and bulge hyper-thickening arise from immune cell stimulation” to clarify our rationale for using Dex rather than ablating specific immune cells to address whether the transcriptomic changes and altered proliferation in HFSCs were rooted in the breached niche or in immune cell infiltration into the breached niche. We have also included the Experimental Dermatology reference, making it clear to the reader that we recognize the potential caveats of Dex but that these are not applicable to our study which focuses on a HF stage that is not known to be affected by Dex.

5) The authors do not perform functional analysis of BMP/FGF signaling in proliferation in their model.

We and others have previously performed functional analyses underscoring the importance of BMP/FGF signaling for stem cell quiescence in the hair follicle, and the importance of inner bulge BMP expression in raising the threshold for stem cell proliferation (Hsu et al., 2011). Figure 6—figure supplement 2A shows the FPKM and qPCR analyses showing that BMP and FGF18 levels are not significantly changed when the inner bulge niche is present, but lacks *Cdh1*. We also show that upon elimination of the immune cell infiltration, pSMAD1, downstream of BMP-signaling still occurs in the HFSCs surrounding a *Cdh1* null niche (Figure 6B). This verifies that the inner bulge niche is still capable of not only expressing *Bmp* genes but also in impacting BMP-signaling in the HFSCs. This differs distinctly from hair plucking, where the inner bulge niche is absent and telogen-phase BMP-signaling is lost (Hsu et al., 2011). That said, the reviewer raises a good point, and we have now added relative expression of the BMP targets *Id1, Id2, Nfatc1* and *Foxc1* in HFSCs and inner bulge cells as revealed by our RNA-seq analysis in Figure 6—figure supplement 2A. Despite the reduction in *Id1* and *Id2* in *Cdh1*-null HFSCs, which is not surprising given their proliferative nature, these cells still maintained expression of the critical HFSC quiescence transcription factors *Nfatc1* and *Foxc1*. Finally, we show that *Id1* and *Id2* are restored in the dexamethasone-treated condition, concomitant with BMP/pSMAD1 signaling, further substantiating that the telogen signals imposed by the inner bulge niche are still operative in the *Cdh1* null niche (Figures 6B and Figure 6—figure supplement 2A).

6) Does the proliferation that is induced in the bulge regenerate the barrier function in the cKO mice?

The reviewer raises an excellent question. Our model is a genetic one, and hence the stem cell’s efforts to patch the barrier are futile. In this regard, it is intriguing that in genetic mutations that affect the suprabasal epidermis, e.g. K10/K1 mutations, for instance, so too is the effort futile, although there the basal progenitors hyper-proliferate too (Fuchs et al., 1992). There is also a fascinating difference between the two stem cell responses: in epidermolytic hyperkeratosis patients/mice (K1/K10 mutations), the proliferative epidermal stem cells generate excessive layers of differentiating cells. In the HF, the proliferative stem cells generate excess stem cell layers but do not add to the defective inner bulge layers. There is an interesting reason for this: in contrast to the epidermis, whose basal progenitors give rise directly to the terminal differentiating cells, in the HF, the stem cells don’t directly differentiate into inner bulge cells; rather the inner bulge layer forms at the end of the hair cycle (Hsu et al., 2011). We have now discussed this intriguing point in the Discussion section entitled “Talking back: immune cell participation in containing the barrier breach”.

As a final note, we planned to perform Lucifer yellow experiments on WT skin after its hairs have been plucked. In this situation, we expected to create a barrier breach under conditions when the HFSCs are only one layer. However, we encountered difficulties in handling plucked HFs and accurately assaying for Lucifer yellow penetration because of their highly distorted morphology upon plucking. Given the caveat introduced by this distortion, the comparison is not a fair one against *Cdh1*-null HFs, whose morphology, besides their hyper-thickened bulge and altered actin network, is still intact.

7) In the figure legends the authors mention n=4 mice. Is that 4 mice per genotype or 2 mice per genotype?

4 mice per genotype – we have revised all figure legends to clarify this point.

Reviewer #2:In the manuscript Lay et al. discover an important interplay between HFSC and immune cells following breaches in the inner bulge niche. Using several strategies, they demonstrate that E-cadherin is important for maintaining adherens junctions, tight junctions, and integrity of the niche, but does not affect niche cell numbers in the bulge. Ablation of Cdh1 specifically in bulge stem cells did not cause the stem cell proliferation phenotype indicating that niche signaling is responsible for stem cell proliferation and expansion of bulge layers in Cdh1 KO mice. Treating Cdh1 KO mice with various antibiotics did not prevent stem cell proliferation, demonstrating that bacterial infiltration is not responsible for stem cell proliferation following niche breach. Dendritic cells (DCs) and αβ T cells, but not macrophages or γδTCR, showed marked and significant recruitment to breached stem cell niches. Immune cell recruitment was mediated by an upregulation of chemokines in Cdh1 KO HFSC and WT HFSC near Cdh1 KO inner bulge niche, revealed with RNA-Seq using FACS sorted populations and validated with in vitro migration assays. Ccl1 and Ccl2, established chemokines for DCs, were the top two most highly upregulated genes in the stem cell cohort. Increases in Ccl1 and Ccl2 occurred in Cdh1 KO HFSC in the presence of antibiotics, demonstrating again that the breached niche response is independent of bacterial infiltration. Furthermore, DC recruitment by HFSC is dependent on DC CCR2 expression, the receptor to Ccl2. Lastly, immunosuppression prevented the HF hyper thickened bulge phenotype and HFSC proliferation, revealing a role for immune cell infiltration in breached niche protection independent of bacterial infiltration. The experiments are controlled well, observations validated using several strategies, and conclusions strongly supported by the data. The manuscript will be of interest to a general audience and is highly novel. I have a few remaining questions.1) Does Ccl1 and Ccl2 expression increase in WT HSFC surrounding Cdh1 KO inner bulge in antibiotic conditions?

In reading this comment as well as the reviewer’s summary statement above, “Increases in *Ccl1* and *Ccl2* occurred in *Cdh1* KO HFSC in the presence of antibiotics, demonstrating again that the breached niche response is independent of bacterial infiltration,” we realized that we have presented our data in a way that is confusing. In fact, *Ccl1* and *Ccl2* are increased both in the WT and in *Cdh1* KO HFSCs that surround a breached niche. These data are shown in Figure 5G. Antibiotics do not abrogate this increase, as we show for the high tamoxifen situation (Figure 5H). The confusion arose from the way we presented the data in Figure 5H. The key point is not the comparison between control HFSCs + antibiotics and *Cdh1* null HFSCs + antibiotics (i.e. purple vs. orange bars), but rather *Cdh1* null HFSCs (in a breached niche) whose skins were either treated (orange bars) or not (red bars) with antibiotics. The conclusion is that in the presence of antibiotics under conditions that prevent the bacterial invasion, the stem cells still express *Ccl1* and *Ccl2* when they experience an inner niche defect and not when they don’t.

We now clarify that (1) the *Cdh1*-null HFSCs in Figure 5H were in a mutant niche and (2) reorganized the bars in a way that guides the reader, i.e. red and orange, followed by blue and purple. Although we cannot combine the *Ccl1* and *Ccl2* data into Figure 5G and H, as one is transcriptome and the other qPCR, we have made it clear to readers that the two sets of data complement each other. The readers will then be able to readily see that irrespective of whether mutant or wild-type, HFSCs that experience a mutant niche specifically upregulate *Ccl1* and *Ccl2*, and that barring bacterial infiltration does not abrogate their expression. We are grateful to the reviewer for their careful analyses of our data and are thankful for the comment.

We have now revised Figure 5H and its accompanying legend to show that *Ccl1* and *Ccl2* are still highly upregulated independent of antibiotics (and hence bacterial infiltration) as there is no statistically significant difference between the red (*Cdh1*-null HFSCs) and red candy-striped (*Cdh1*-null HFSCs with antibiotics) bars.

2) Why is an increase in HFSC Ccl1 important? Ccl1 interacts with CCR8, expressed by DCs, T cells and monocytes. Were in vitro migration assays using CCR8 KO DCs or T cells performed?

The reviewer raises an interesting point. The reason why we pursued only CCL2 and not also CCL1 is because the CCR8 mice are not commercially available in contrast to CCL2 null mice. When we performed the tests on DC migrations in WT versus CCR2 mutant backgrounds, we saw that the migratory effect was diminished, underscoring CCR2-CCL2’s role. That said, CCL1-CCR8 interactions could still play a role as the migration was compromised but not completely blocked. We have now performed in vitro migration assays using CCR8 blocking antibody. We found that blocking the CCL1-CCR8 axis had a less profound effect on DC and macrophage migration (new Figure 5J). However, treating CCR2-KO DCs and macrophages with CCR8 blocking antibody to block both CCL2-CCR2 and CCL1-CCR8 signaling had an additive effect, dampening DC and macrophage migration more than blocking either pathway alone (updated Figure 5I). As the reviewer suggested, we have also performed migration assay for Tregs. However, we did not observe a significant difference in Treg migration induced by *Cdh1* cKO HFSCs compared to WT HFSCs. Therefore, we did not perform additional treatment of Tregs with CCR8 blocking antibody.

Note that we contemplated but do not think it is a valid approach to try performing CCL1 and CCL2 shRNA knockdown experiments on our purified HFSCs, since we are isolating HFSCs fresh from a breached niche, and the time for us to carry out and confirm efficient shRNA knockdown could pose a new caveat.